# Deconstructing Intratumoral Heterogeneity through Multiomic and Multiscale Analysis of Serial Sections

**DOI:** 10.3390/cancers16132429

**Published:** 2024-07-01

**Authors:** Patrick G. Schupp, Samuel J. Shelton, Daniel J. Brody, Rebecca Eliscu, Brett E. Johnson, Tali Mazor, Kevin W. Kelley, Matthew B. Potts, Michael W. McDermott, Eric J. Huang, Daniel A. Lim, Russell O. Pieper, Mitchel S. Berger, Joseph F. Costello, Joanna J. Phillips, Michael C. Oldham

**Affiliations:** 1Department of Neurological Surgery, University of California, San Francisco, San Francisco, CA 94143, USA; pschupp@sonic.net (P.G.S.); sam_shelton@hotmail.com (S.J.S.); daniel.brody@ucsf.edu (D.J.B.); rebecca.eliscu@ucsf.edu (R.E.); johnbret@ohsu.edu (B.E.J.); tmazor@ds.dfci.harvard.edu (T.M.); kwkelley@stanford.edu (K.W.K.); matthew.potts@nm.org (M.B.P.); mwmcd@baptisthealth.net (M.W.M.); daniel.lim@ucsf.edu (D.A.L.); russ.pieper@ucsf.edu (R.O.P.); mitchel.berger@ucsf.edu (M.S.B.); joseph.costello@ucsf.edu (J.F.C.); joanna.phillips@ucsf.edu (J.J.P.); 2Biomedical Sciences Graduate Program, University of California, San Francisco, San Francisco, CA 94143, USA; 3Medical Scientist Training Program, University of California, San Francisco, San Francisco, CA 94143, USA; 4Neuroscience Graduate Program, University of California, San Francisco, San Francisco, CA 94143, USA; 5Department of Pathology, University of California, San Francisco, San Francisco, CA 94143, USA; eric.huang2@ucsf.edu

**Keywords:** intratumoral heterogeneity, clonal evolution, tumor microenvironment, low-grade glioma, IDH1, gene coexpression, multiomic, single-nucleus analysis

## Abstract

**Simple Summary:**

Tumors contain cancerous cells with different mutations and different types of non-cancerous cells. It is important to understand the molecular properties of these cells in order to develop effective treatments for cancers. We describe a new strategy for studying tumor heterogeneity and apply it to human astrocytomas, a type of brain tumor. By slicing astrocytomas into a large number of serial sections and analyzing patterns of mutation frequencies, we reconstruct the evolutionary history of cancer cell mutations in each tumor. By comparing these patterns to gene activity measured in the same sections, we identify molecular features that optimally distinguish different types of cancerous and non-cancerous cells and validate these with different techniques, including analysis of individual nuclei. By integrating analyses of multiple molecular species at multiple scales, our strategy provides a powerful approach for precisely deconstructing intratumoral heterogeneity.

**Abstract:**

Tumors may contain billions of cells, including distinct malignant clones and nonmalignant cell types. Clarifying the evolutionary histories, prevalence, and defining molecular features of these cells is essential for improving clinical outcomes, since intratumoral heterogeneity provides fuel for acquired resistance to targeted therapies. Here we present a statistically motivated strategy for deconstructing intratumoral heterogeneity through multiomic and multiscale analysis of serial tumor sections (MOMA). By combining deep sampling of IDH-mutant astrocytomas with integrative analysis of single-nucleotide variants, copy-number variants, and gene expression, we reconstruct and validate the phylogenies, spatial distributions, and transcriptional profiles of distinct malignant clones. By genotyping nuclei analyzed by single-nucleus RNA-seq for truncal mutations, we further show that commonly used algorithms for identifying cancer cells from single-cell transcriptomes may be inaccurate. We also demonstrate that correlating gene expression with tumor purity in bulk samples can reveal optimal markers of malignant cells and use this approach to identify a core set of genes that are consistently expressed by astrocytoma truncal clones, including *AKR1C3*, whose expression is associated with poor outcomes in several types of cancer. In summary, MOMA provides a robust and flexible strategy for precisely deconstructing intratumoral heterogeneity and clarifying the core molecular properties of distinct cellular populations in solid tumors.

## 1. Introduction

Tumors are complex ecosystems containing huge numbers of malignant and nonmalignant cells. Malignant cells evolve over time by acquiring mutations through diverse mechanisms that promote genetic [1] and epigenetic [2] heterogeneity, which may occur in a neutral fashion [3] or as a Darwinian response to therapeutic or other environmental pressures [4]. Nonmalignant cells comprise diverse tumor microenvironments (TMEs) that vary within and among tissues and individuals and may be influenced by malignant cells to adopt tumor-suppressive or tumor-supportive behaviors [5,6,7]. The genetic, epigenetic, and microenvironmental diversity of individual tumors is collectively described as intratumoral heterogeneity (ITH) [8]. Clarifying the extent of ITH is an important goal for precision medicine, since most mutations are not shared between malignant clones from different individuals [9,10,11,12,13], and ITH provides the substrate for acquired resistance to targeted therapies [8,14,15,16].

Investigators have mostly studied ITH by applying multiomic assays to a small number of bulk subsamples from the same tumor. Multi-region analyses of renal carcinoma [17], breast cancer [18], colorectal cancer [19], glioblastoma [20], and others [21] have identified spatial variation in mutation frequencies and other molecular phenotypes, revealing extensive ITH. However, this experimental design is high-dimensional in omics feature space but low-dimensional in sample space, which can lead to biased inference and inflated false-positive error rates for molecular features [22]. Furthermore, the small number of bulk samples limits any conclusions that can be drawn about distinct malignant clones and nonmalignant cell types. Recent efforts using single-cell methods have provided new perspectives on ITH [23,24,25], but it remains non-trivial to isolate and sequence DNA and RNA from the same cell at scale. As such, cancer cells are often identified from copy-number variants (CNVs) inferred from single-cell data. However, single-cell data are confounded by technical factors related to tissue dissociation, sampling bias, noise, contamination, and sparsity [26,27,28,29,30], which muddle the relationships between malignant cell genotypes and molecular phenotypes, particularly for cancers that lack consistent CNVs.

We have shown that variation in the cellular composition of intact human brain samples drives the covariation of transcripts that are uniquely or predominantly expressed in specific kinds of cells [31,32,33,34]. We have also shown that the correlation between a gene’s expression pattern and the abundance of a cell type is a proxy for the extent to which the same gene is differentially expressed by that cell type [31]. These findings suggest that the core molecular properties of malignant clones can be identified in bulk tumor samples by correlating molecular feature levels with clonal abundance, which can be quantified through an integrative analysis of variant allele frequencies (VAFs) [35,36]. In principle, such findings should be highly robust since they derive from millions or even billions of cells. A similar logic can be extended to nonmalignant cell types of the TME [31].

Here we describe a novel strategy for deconstructing ITH through multiomic and multiscale analysis (MOMA) of serial tumor sections. By analyzing gene expression, whole exomes, deeply sequenced PCR amplicons spanning mutation sites, DNA methylation, and single-nucleus DNA and RNA, we exhaustively analyze ITH in IDH-mutant astrocytomas. Through integrative analysis of single-nucleotide variants (SNVs) and CNVs, we precisely define the evolutionary histories and spatial distributions of malignant clones. By comparing these distributions to gene expression data derived from the same tumor sections, we reveal clone-specific transcriptional profiles and validate them orthogonally through comparisons with normal human brains and single-nucleus analysis. Our findings suggest that a core set of genes are consistently expressed by the truncal clone of human astrocytomas, suggesting new therapeutic targets and a generalizable strategy for precisely deconstructing ITH and clarifying the core molecular properties of distinct cellular populations in solid tumors.

## 2. Results

### 2.1. Overview of MOMA

Figure 1a depicts a heterogeneous human brain tumor specimen consisting of distinct malignant clones and nonmalignant cell types of the TME. By amplifying this specimen into a series of standardized biological replicates through serial sectioning, we introduce variation in cellular composition across sections (Figure 1b), which are analyzed using multiscale (bulk and single-nucleus) and multiomic assays (Figure 1c). Correlative analysis of cellular frequencies and molecular feature levels (e.g., gene expression levels) in bulk sections predicts optimal markers of distinct malignant clones and nonmalignant cell types (Figure 1d), which are validated by single-nucleus analysis of interpolated sections and histology (Figure 1e). MOMA therefore combines the power of bulk sampling with the precision of single-cell analysis to achieve the best of both worlds.

### 2.2. Case 1: Analysis of Clonal Composition

To put these ideas into practice, we obtained a resected specimen from a primary diffuse glioma that was removed from the left cerebral hemisphere of a 40 y.o. female who presented with language deficits (Figure 2a–c). Its molecular pathology revealed evidence of mutations in *IDH1* and *TP53* (Figure 2d,e), no evidence for chromosome 1p/19q codeletion, and a KI67 labeling index of 6%, consistent with a CNS WHO grade 2 IDH-mutant astrocytoma. We cut 81 cryosections along the tumor specimen’s longest axis (Figure 2f), followed by automated DNA/RNA extraction from each section (Appendix A). To identify mutations and characterize the clonal landscape, we performed whole-exome sequencing (WES) on DNA isolated from sections 14, 39, and 69 and the patient’s blood. Mutations detected in blood or in genes with very low tumor expression levels were excluded. Of the remaining 33 mutations (Appendix A), including an in-frame deletion in *ATRX*, which is often mutated in IDH-mutant astrocytomas [37], 18 were validated by Sanger sequencing and deep sequencing of PCR amplicons spanning each mutation (amp-seq; Appendix A), five were validated only by amp-seq, and ten (mostly indels) could not be validated (Appendix A). Among 22 validated coding mutations, 16 were detected by WES in all three tumor sections, and six were detected in only one section, suggesting clonal heterogeneity among malignant cells (Appendix A).

To determine the relative abundance and spatial distributions of cells carrying mutations within the tumor specimen, we quantified VAFs for validated somatic mutations in all tumor sections using amp-seq (Appendix A). Amp-seq was performed in two sequencing runs: an initial run consisting of 25 amplicons (mean coverage: 3.0 × 10^3^ reads/mutation/section) and a second run consisting of nine amplicons (mean coverage: 1.7 × 10^4^ reads/mutation/section), with a theoretical VAF detection sensitivity of <1%. To analyze the stability of amp-seq-derived VAFs, we downsampled reads spanning IDH1 R132H or TP53 L145P and calculated the root mean square error (RMSE) and Pearson correlation between VAFs from full and downsampled read depths. This analysis revealed a monotonic improvement in VAF estimates as a function of the read depth (Appendix A). Importantly, VAFs derived from 100–200× coverage were far noisier than VAFs derived from full coverage, indicating that conventional WES data are inadequate for precisely estimating VAFs and malignant cell abundance.

We performed unsupervised hierarchical clustering of amp-seq data to identify mutations with similar VAF patterns within the tumor sample (Figure 2g and Appendix A). This analysis revealed three distinct clusters. Cluster 1 included 15 mutations with VAFs that decreased in the latter sections of the tumor sample, which were separated according to sequencing run to display the effects of read depth (Figure 2h,i). Cluster 2 included four mutations with VAFs that increased in the latter sections of the tumor sample (Figure 2j). Cluster 3 included three mutations with VAFs that peaked in the middle sections of the tumor sample (Figure 2k).

Focusing on the sequencing run with higher coverage, we observed that five mutations in cluster 1 (including IDH1 R132H) had VAFs over all tumor sections that were statistically indistinguishable (Figure 2h). Two other mutations (TP53 L145P and ACCS A197T) followed a similar pattern but at different scales. qPCR testing revealed approximately disomic copy numbers for both genes in all analyzed sections (Appendix A). These results indicate that VAFs for TP53 L145P reflect a copy-neutral loss of heterozygosity for chromosome 17p (chr17p LOH), which occurred early in the tumor’s evolution (but after the L145P point mutation). Notably, the frequencies of chr17p LOH (derived from B-allele frequencies) were highly concordant between the WES and amp-seq data (r = 0.99, Appendix A (top)). In contrast, the lower VAFs for ACCS A197T suggest that this mutation appeared after the other mutations comprising cluster 1.

To determine the clonal composition and evolutionary history of the tumor specimen more precisely, we analyzed genome-wide CNVs and their relationships to SNVs quantified by amp-seq. CNVs were called from WES (n = three sections) and DNA methylation (n = 68 sections) data using FACETS [38] and ChAMPS [39], respectively, yielding highly concordant frequencies for copy number changes (r = 0.92, Appendix A (bottom) and Appendix A). Through combined analysis of SNV and CNV frequencies over all tumor sections, we produced an integrated model of tumor evolution. Specifically, we used PyClone [35] to jointly analyze SNV and CNV frequencies, which identified seven distinct clusters and their overall prevalence. Subsequently, the evolutionary history of the tumor specimen was reconstructed using CITUP [36], which produced the most likely phylogenetic tree (Figure 2l) and frequencies of six malignant clones over all sections (Figure 2m and Appendix A). These analyses confirmed the truncal nature of mutations in *IDH1* and *TP53* [37], while revealing a wide variation in the purity of individual tumor sections (range: 38.3–84.8%; Appendix A).

### 2.3. Case 1: Analysis of Gene Expression

We next explored relationships between clonal abundance and gene expression. We performed genome-wide gene coexpression analysis to identify groups of genes with similar expression patterns over all tumor sections. We identified 38 modules of coexpressed genes (arbitrarily labeled by colors), which were summarized by their eigengenes (i.e., first principal components) and hierarchically clustered (Appendix A, Figure 3a–c). As we have shown previously [31,32,33,34], many modules were significantly enriched with markers of nonmalignant cell types (Appendix A). By comparing cumulative clonal abundance (Figure 2m) to module eigengenes over all tumor sections, we identified five gene coexpression modules whose expression patterns closely tracked the abundance of clone 1 (turquoise: r = 0.97, Figure 3d), clone 3 (blue: r = 0.84, Figure 3e), clone 4 (black: r = 0.83, Figure 3f), clone 5 (midnightblue: r = 0.71, Figure 3g), and clone 6 (steelblue: r = 0.69). We did not identify a module that was significantly correlated with clone 2, which represented only 0.001% of cells (Figure 2l).

To characterize these modules, we performed enrichment analysis with biologically relevant gene sets (Figure 3d–g). We first asked whether genes within clonal CNV boundaries (Figure 2l and Appendix A) were significantly enriched (for gains) or depleted (for deletions) in the bulk coexpression modules most strongly associated with each clone (Appendix A). Notably, all such gene sets were significantly enriched in the appropriate module and expected direction (e.g., chr7 gain for clone 1 (Figure 3d), chr2p deletion for clone 3 (Figure 3e), and chr10p gain for clone 5 (Figure 3g)). We next analyzed publicly available gene sets from diverse sources (Appendix A). We found that the largest (turquoise) module, which closely tracked the abundance of clone 1 (i.e., tumor purity), was significantly enriched with markers of oligodendrocyte progenitor cells (OPCs) and radial glia, genes comprising the ‘classical’ subtype of glioblastoma proposed by Verhaak et al. [40] and numerous gene sets related to microglial infiltration and activation. The second largest (blue) module, which tracked clone 3, was significantly enriched with neuronal gene sets as well as genes that are downregulated pursuant to *IDH1* mutations. The black module, which tracked clone 4, was enriched with astrocyte markers as well as genes that are differentially regulated during development and glioma. The midnightblue module, which tracked clone 5, was enriched with markers of smooth muscle cells, genes comprising the ‘mesenchymal’ subtype of glioblastoma [40,41], and gene sets related to the epithelial–mesenchymal transition and invasiveness. The steelblue module, which tracked clone 6, was enriched with markers of non-resident immune cells.

To further characterize the transcriptional signatures associated with each clone, we used multiple linear regression to model genome-wide expression levels as a function of clonal abundance. To account for collinearity and the dominant effect of clone 1, we used a group lasso model with bootstrapped clonal abundance vectors (real or permuted) as predictors (Appendix A). We restricted our focus to genes that were significantly and stably modeled by a single clone (in addition to clone 1, per the group lasso model, Appendix A). The enrichment analysis of these genes largely recapitulated the enrichment analysis of gene coexpression modules associated with each clone, including the associations of different clones with different cell types (Appendix A and Appendix A).

The associations of different clones with different cell types suggest two non-mutually exclusive possibilities. First, different clones may preferentially express different cell-type-specific transcriptional programs. Second, different clones may preferentially associate with different nonmalignant cell types in the TME, leading to correlated gene expression patterns. Although such possibilities are ideally studied at the level of individual cells, all sections from this case were consumed during bulk data production. However, we reasoned that bona fide transcriptional signatures of malignant clones should be absent from non-neoplastic human brains. To test this hypothesis, we profiled gene expression in 361 cryosections from four neurotypical adult human brain samples (Appendix A) and performed a genome-wide differential coexpression analysis by subtracting normal correlations from tumor correlations, such that tumor-specific gene coexpression relations were retained (Appendix A). This analysis revealed tumor-specific gene coexpression modules that tracked the abundance of distinct clones and largely recapitulated the transcriptional signatures described in Figure 3 and Appendix A, including the preserved enrichment of clone-specific CNV gene sets (Appendix A). However, the enrichment results for nonmalignant cell-type-specific gene sets became less significant, with the exception of OPCs and radial glia for clone 1, which became more significant (Appendix A). These results suggest that derived clones may occupy distinct microenvironments, while the truncal clone retains signatures of progenitor cells that may reflect the cell of origin.

### 2.4. Case 2: Analysis of Clonal Composition

To test our strategy on a more complex case, we obtained a resected specimen from a recurrent diffuse glioma that was removed from the right cerebral hemisphere of a 58 y.o. male (Figure 4a–c) approximately 28 years after the primary resection. Its molecular pathology revealed evidence of mutations in *IDH1* and *TP53* (Figure 4d–e), no evidence of chromosome 1p/19q codeletion, and a KI67 labeling index of 4%, consistent with a recurrent CNS WHO grade 2 IDH-mutant astrocytoma. Building on our observations from case 1, we applied the same strategy to case 2, with five modifications. First, we increased the level of power by analyzing more sections (Appendix A). Second, we rotated the sample 90° halfway through sectioning to capture ITH in orthogonal planes (Figure 4f). Third, we inferred CNVs from RNA-seq data instead of DNA methylation data. Fourth, we increased the average sequencing depth for amp-seq data. And fifth, we analyzed single nuclei from interpolated sections to validate predictions from bulk sections (Figure 4f).

To identify somatic mutations, we performed WES on DNA from two sections in each plane (22, 46, 85, 123; Appendix A) and the patient’s blood. A total of 227 mutations were identified, and 74 were selected for amp-seq by clustering WES VAFs to reveal candidate mutations most likely to mark distinct clones (Appendix A). Of these, 58 mutations were verified by amp-seq (Appendix A). As with case 1, downsampling reads spanning IDH1 R132H or TP53 G245V revealed monotonic improvements in VAF estimates as a function of read depth (Appendix A). We therefore restricted further analysis of amp-seq data to 27 mutations with high coverage over all tumor sections or strong VAF correlations to other mutations (Appendix A). Hierarchical clustering of these amp-seq data (Appendix A) revealed five clusters of mutations with similar VAF patterns within the tumor sample (Figure 4g–l), suggesting multiple malignant clones.

Because mutations in *IDH1*, *TP53*, and *ATRX* are considered diagnostic for astrocytoma [37], we expected these to be truncal and were therefore surprised that IDH1 R132H fell in a separate cluster from mutations in *TP53* and *ATRX* (Figure 4j,k). To explore this discrepancy, we analyzed VAFs for all three mutations after controlling for gene dosage. This analysis revealed a greater discordance between VAFs for *IDH1* and *TP53/ATRX* mutations in sectioning plane 1 vs. sectioning plane 2 (Figure 4m). We also observed that all genes in mutation cluster 4 (including *IDH1*) are located on chr2q. These observations suggested that the discrepancy between *IDH1* and *TP53*/*ATRX* mutation VAFs might be explained by a subclonal deletion in chr2q pursuant to the IDH1 R132H mutation, as has been previously reported [42,43,44]. To test this hypothesis, we quantified CNVs from WES (n = four sections) and RNA-seq (n = 90 sections) data using FACETS [38] and CNVkit [45], respectively, which yielded highly concordant frequencies for copy number changes (r = 0.97, Appendix A and Appendix A), including a chr2q deletion event. As expected, frequencies of the chr2q deletion event were substantially higher in sectioning plane 1 vs. sectioning plane 2 (Figure 4n) and almost perfectly correlated with the observed discordance between *IDH1* and *TP53*/*ATRX* mutation VAFs (r = 0.98, Appendix A).

Through combined analysis of SNV and CNV frequencies over all tumor sections, we generated an integrated model of tumor evolution using the same approach as that described for case 1, including the most likely phylogenetic tree (Figure 4o) and frequencies of five malignant clones over all sections (Figure 4p and Appendix A). Compared to case 1, there was substantially less variation in the purity of individual tumor sections (range: 71.4–81.6%; Appendix A). We confirmed the truncal nature of mutations in *IDH1*, *TP53*, and *ATRX*, along with gains of chr7, chr8, and chr9. To more closely examine the sequence of early mutational events, we performed single-nucleus DNA sequencing using MissionBio’s Tapestri microfluidics platform [46]. We took advantage of an existing panel of cancer genes, which included primers flanking one *IDH1* and two *TP53* loci. We were also able to infer chr17 and chr2q copy-number changes using mutations that fell within the targeting panel. We analyzed 4433 nuclei from plane 1 (section 29) and 3736 nuclei from plane 2 (sections 113 and 115). Clustering the nuclei from each plane revealed clonal frequencies that broadly matched those obtained by bulk analysis (Appendix A, Appendix A). Interestingly, we observed a subpopulation of clone 1 (clone 1a: 4.1–6.6%) with IDH1 R132H −/+ and TP53 G245V −/+/+ genotypes (Appendix A). These genotypes suggest that *TP53* LOH occurred mechanistically in this case through the duplication of the mutant allele prior to loss of the wild-type allele and may also explain the slightly lower VAFs for TP53 G245V compared to those of the mutation in *ATRX* (Figure 4j).

### 2.5. Case 2: Analysis of Gene Expression

We explored relationships between clonal abundance and gene expression using the same strategies described for case 1. Genome-wide gene coexpression analysis identified 68 modules of coexpressed genes, which were summarized by their eigengenes and hierarchically clustered (Figure 5a–c). As expected [31,32], many modules were significantly enriched with markers of nonmalignant cell types (Appendix A). By comparing clonal abundance (Figure 4p, Appendix A) to module eigengenes over all tumor sections, we identified five gene coexpression modules whose expression patterns closely tracked the abundance of clone 1 (red: r = 0.65, Figure 5d), clone 2 (violet: r = 0.82, Figure 5e), clone 3 (black: r = 0.8, Figure 5f), clone 4 (ivory: r = 0.86, Figure 5g), and clone 5 (lightcyan: r = 0.82).

Enrichment analysis using gene sets defined by clonal CNV boundaries (Figure 4o and Appendix A) confirmed the expected over-representation (for gains) or under-representation (for deletions) in the bulk coexpression modules most strongly associated with each clone (Figure 5d–g, Appendix A). Further analysis using publicly available gene sets from diverse sources (Appendix A) revealed that the red module, which tracked the abundance of clone 1 (i.e., tumor purity), was significantly enriched with markers of radial glia and microglia, as well as genes comprising the mesenchymal subtype of glioblastoma. The violet module, which closely tracked the abundance of clone 2, was significantly enriched with genes from reported astrocytoma expression programs, as well as TNFalpha signaling and extracellular matrix components. The black module, which closely tracked the abundance of clone 3, was significantly enriched with markers of neurons and genes involved in chromatin remodeling. The ivory module, which closely tracked the abundance of clone 4, was enriched with markers of ependymal cells and myeloid cells. The lightcyan module, which closely tracked the abundance of clone 5, was significantly enriched with genes involved in EGFR and NF-kB signaling, as well as genes comprising the proneural subtype of glioblastoma.

To further characterize the transcriptional signatures associated with each clone, we used multiple linear regression to model genome-wide expression levels as a function of clonal abundance. To account for collinearity, we used a regular lasso model with bootstrapped clonal abundance vectors (real or permuted) as predictors (Appendix A). We restricted our focus to genes that were significantly and stably modeled by a single clone (Appendix A). Enrichment analysis of these genes largely recapitulated the enrichment analysis of gene coexpression modules associated with each clone, including CNVs and the associations of different clones with different cell types (Appendix A, Appendix A).

To validate gene expression signatures of malignant clones and nonmalignant cell types identified from bulk tumor sections, we performed single-nucleus RNA-seq (snRNA-seq) on tumor sections 17, 53, 93, and 117 (Figure 4f, Appendix A). Using a protocol adapted from TARGET-Seq [47,48], we profiled gene expression in 288 flow-sorted nuclei per section. Following data preprocessing and quality control, 809 nuclei (70.2%) with an average of >200K unique reads/nucleus were retained for further analysis. Uniform manifold approximation and projection (UMAP) analysis revealed that the nuclei did not segregate by section ID (Appendix A, Appendix A).

To determine whether the nuclei segregated by cancerous state, we analyzed the malignancy of each nucleus. Unlike some tumors, astrocytomas are not defined by truncal CNVs, which can drive gene expression changes that are used to infer malignancy in snRNA-seq data [25,37,49,50]. We therefore genotyped all 809 nuclei through single-nucleus amplicon sequencing (snAmp-seq) of cDNA spanning mutations in the truncal clone (Figure 4o). This analysis provided sufficient information to assign malignant status for 75% of the nuclei. Projecting malignancy status onto the UMAP plot revealed clear segregation of malignant and nonmalignant nuclei (Appendix A).

To further classify the nuclei as specific malignant clones or nonmalignant cell types, we took a two-step approach. First, we hierarchically clustered all of the nuclei using a Bayesian distance metric calculated by Sanity [27] that downweights genes with large error bars, revealing 12 clusters. Second, we asked whether the genes in the bulk coexpression modules that were most strongly associated with each malignant clone or nonmalignant cell type were upregulated in distinct snRNA-seq clusters compared to all other genes (Appendix A). This analysis revealed specific and significant upregulation of genes from the red (Figure 5d), violet (Figure 5e), black (Figure 5f), and lightcyan modules in snRNA-seq clusters 2, 1, 7, and 10 (Figure 6a), suggesting that these clusters correspond to malignant clones 1, 2, 3, and 5, respectively. Genes in the ivory module (Figure 5g) were significantly upregulated in snRNA-seq clusters 3 and 5, suggesting that both of these clusters represent clone 4 (Figure 6a). Similarly, we observed specific and significant upregulation of genes from the purple (Appendix A), yellow (Appendix A), green (Appendix A), and orange modules in snRNA-seq clusters 9, 4, 12, and 6 (Figure 6a), suggesting that these clusters correspond to nonmalignant astrocytes, microglia, neurons, and endothelial cells, respectively. Genes in the tan module (Appendix A) were significantly upregulated in snRNA-seq clusters 8 and 11, suggesting that both of these clusters represent nonmalignant oligodendrocytes (Figure 6a). All *p*-values and effect sizes are reported in Appendix A.

We performed several additional analyses to verify these findings. First, we projected snRNA-seq cluster assignments onto the UMAP plot (Figure 6b) and observed that the cluster assignments were consistent with the malignancy map produced by genotyping the nuclei via snAmp-seq (Appendix A). Second, we performed UMAP analysis for the malignant cells only, followed by trajectory analysis with Slingshot [51] (Figure 6c). This analysis revealed patterns of clonal evolution that recapitulated the phylogenetic tree inferred from integrative analysis of bulk tumor sections (Figure 4o). Third, we compared estimates of cellular abundance obtained from the bulk and single-nucleus data for adjacent tissue sections. This analysis revealed highly consistent estimates for the relative abundance of malignant clones (r ≥ 0.94; Appendix A) and nonmalignant cell types (r ≥ 0.90; Appendix A).

Supervised clustering with differentially expressed genes revealed a clear separation of snRNA-seq clusters (Figure 7). Overall, malignant clones were more transcriptionally active than nonmalignant cell types, with the exceptions of clone 4:1 and endothelial cells (Figure 7, right). Enrichment analysis of genes that were significantly upregulated in the snRNA-seq clusters confirmed the identities of nonmalignant cell types (Appendix A, Appendix A). For malignant clones, enrichment analysis of the snRNA-seq clusters supported and refined our inferences from the bulk data (Figure 5d–g, Appendix A, Appendix A). For clone 1, consistent enrichments for markers of radial glia and genes comprising the mesenchymal subtype of glioblastoma were observed in the bulk and snRNA-seq data. In contrast, markers of microglia were less significantly enriched in clone 1 nuclei from snRNA-seq data versus bulk data, and markers of oligodendrocyte progenitor cells (OPCs) were more significantly enriched. For clone 2, markers of astrocytes were more significantly enriched in snRNA-seq data versus bulk data. Clone 3 was consistently enriched with genes involved in chromatin remodeling, but neuronal markers were less significantly enriched in snRNA-seq data. Clone 4 showed strong enrichment for markers of ependymal cells in all analyses, while clone 5 was significantly enriched with genes comprising the proneural subtype of glioblastoma in all analyses. Interestingly, genes involved in mitosis were most highly expressed by clone 1, clone 4:2, and endothelial cells (Figure 7, right).

Because the clones in this case were characterized by disparate CNVs (Figure 4o), we asked how malignancy calls compared between algorithms that infer CNVs from snRNA-seq data and malignant genotypes derived from the snAmp-seq data. We used CopyKat [49], InferCNV [25], and CaSpER [50] to call CNVs from the snRNA-seq data. These analyses revealed substantial variation in malignancy calls for different algorithms (Figure 7) as well as differences from bulk CNV calls (e.g., no gains in chr7p, chr8p, and chr9q; Appendix A). Taking the snAmp-seq genotyping as ground truth, CopyKat and InferCNV were more sensitive but less specific than CaSpER, leading to discrepant calls. For example, nonmalignant astrocytes and oligodendrocytes:2 were mostly called malignant by CopyKat and InferCNV, while clone 4:2 was mostly called nonmalignant by these two algorithms. CaSpER’s classification of nuclei from these populations was mostly correct, but it failed to recognize most malignant nuclei for clones 3 and 5. In addition, clone 4:1 was mostly classified as nonmalignant by all three algorithms. Overall, none of the algorithms for inferring malignancy from CNVs achieved accuracy > 61% (Figure 7).

### 2.6. Integrative Analysis of Gene Expression in Malignant Cells

Intuitively, genes whose expression patterns correlate most strongly with the abundance of malignant cells should include optimal biomarkers. This intuition can also be proven mathematically and empirically. Figure 8a–c illustrates a hypothetical example in which the goal is to identify optimal transcriptional markers of malignant cells in a human brain tumor. A conventional strategy would involve physically isolating individual cells, transcriptionally profiling them by single-cell RNA-seq (scRNA-seq), inferring the malignancy of individual cells from the scRNA-seq data based on the presence of driver mutations (CNVs and/or SNVs), and performing a differential expression analysis for each gene between all malignant and nonmalignant cells (for example, using a *t*-test; Figure 8b). Figure 8c shows an alternative analytical path that leads to the same place: by correlating expression levels of the same hypothetical gene from Figure 8b with a dichotomous variable denoting malignant cell abundance (1 = malignant cells, 0 = nonmalignant cells), the resulting statistical significance is identical to that obtained by differential expression analysis.

Although the t-test and correlation produce identical results when the independent variable is dichotomous, this is not the case when the independent variable is continuous. However, we have shown the following via pseudobulk analysis of scRNA-seq data from normal adult human brains: (i) the correlation between the expression pattern of a gene and the (continuous) abundance of a cell type accurately predicts the differential expression of that gene in that cell type; and (ii) cell-type-specific gene coexpression relationships accurately predict cellular abundance in pseudobulk samples [31]. To determine whether these findings extend to malignant cells, we repeated this analysis using scRNA-seq data from 10 adult human astrocytomas [24] (Figure 8d). Genome-wide gene coexpression analysis of pseudobulk samples obtained by randomly aggregating scRNA-seq data revealed a malignant cell coexpression module whose first principal component (or eigengene [52]) closely tracked the actual abundance of sampled malignant cells (Figure 8e,f). Furthermore, the genes that were most significantly upregulated in malignant cells as per the differential expression analysis of the underlying scRNA-seq data (Figure 8d) also had the highest correlations to malignant cell abundance (*k*_ME_) in the pseudobulk data (Figure 8g). These results confirm that optimal markers of malignant cells can be revealed by correlating genome-wide expression patterns with malignant cell abundance in heterogeneous tumor samples. This strategy also applies to individual malignant clones (as well as nonmalignant cell types of the TME), as shown for case 2 in Appendix A.

We next sought to compare the transcriptional profiles of malignant cells between case 1 and case 2 through integrative analysis. However, despite the fact that both tumors were diagnosed as grade 2 IDH-mutant astrocytomas, only one SNV was shared between the cases. Furthermore, the shared SNV (IDH1 R132H) was absent in ~21% of malignant cells in case 2 following the loss of chr2q (Figure 4o). We therefore asked whether the truncal clones (i.e., clone 1), which presumably included all of the mutations required to initiate these tumors along with passenger mutations, had consistent transcriptional profiles in case 1 and case 2. For each case, we analyzed genome-wide correlations to the cumulative abundance of clone 1 (equivalent to the tumor purity). Comparing these results between cases, we observed a highly significant relationship (Figure 9a, Appendix A). The enrichment analysis of genes whose expression patterns were most positively correlated with clone 1 in both cases implicated gene sets comprising the ‘classical’ subtype of glioblastoma proposed by Verhaak et al. [40], markers of radial glia, infiltrating monocytes, and extracellular matrix components (Figure 9a,b; red). In contrast, genes whose expression patterns were most negatively correlated with clone 1 in both cases largely implicated gene sets related to neurons and neuronal function (Figure 9a,b; blue).

We further characterized genes whose expression patterns were most positively correlated with the truncal clone in both cases (Figure 9a; red) by cross-referencing them with human protein–protein interaction (PPI) data from the STRING database [53,54]. This analysis revealed eight distinct clusters of interacting proteins (Figure 9c). The largest of these (green) included several SOX transcription factors and was significantly enriched with genes involved in WNT and MYC signaling (Figure 9d). The second largest cluster (yellow) was significantly enriched with genes involved in DNA repair, and the third largest cluster (orange) was significantly enriched with genes involved in RNA splicing (Figure 9d). The remaining clusters were significantly enriched with genes involved in mRNA transport (brown), DNA replication (turquoise), specific cellular compartments and protein complexes (pink, gray), and immune response (purple) (Figure 9d).

To provide further validation for these findings, we performed immunostaining for AKR1C3. Out of 15,288 genes, *AKR1C3* bulk expression correlations to tumor purity ranked fifth in case 1 and first in case 2 (Figure 9a (asterisk), Appendix A). *AKR1C3* was also significantly upregulated in malignant vs. nonmalignant nuclei per snRNA-seq analysis (Figure 7, right). Immunostaining confirmed the substantial upregulation of AKR1C3 in tumor vs. normal human brain at the protein level (Figure 9e,f). To provide cellular resolution, we co-stained for AKR1C3 and IDH1 R132H using an antibody that recognizes the mutated IDH1 protein. As expected, this analysis revealed broad overlap between cells expressing AKR1C3 and cells expressing IDH1 R132H (Figure 9g–i).

**Figure 9 cancers-16-02429-f009:**
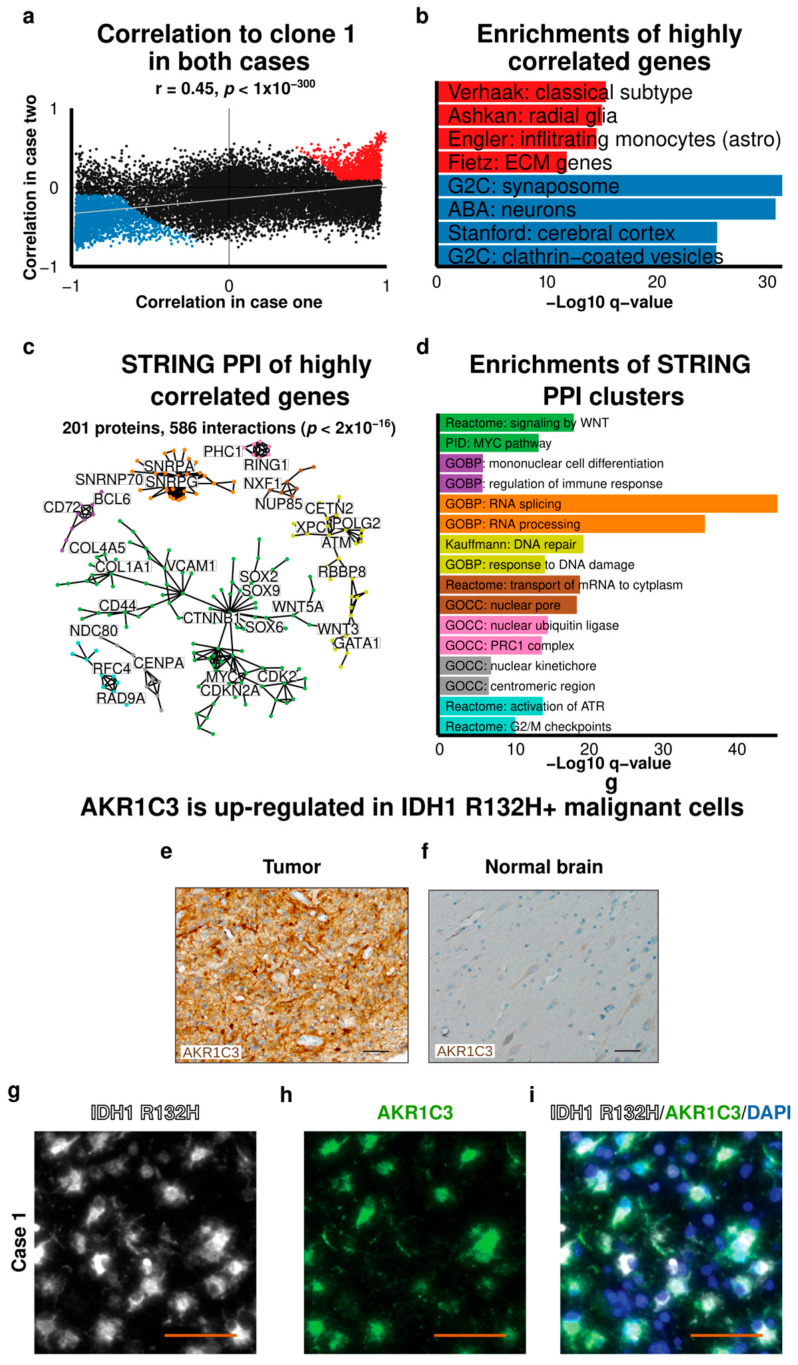
Aggregating correlations to tumor purity reveals core transcriptional features of astrocytomas. (**a**) Gene expression correlations (n = 15,288 genes) to malignant cell abundance in case 1 and case 2. Red and blue denote significantly correlated genes that were used for enrichment analysis (**b**), and the star denotes *AKR1C3*. (**b**) −Log_10_ FDR-corrected *p*-values (q-values) from one-sided Fisher’s exact tests analyzing gene set enrichment in red and blue genes from (**a**). (**c**) Validated protein–protein interactions (PPI) from STRINGdb [54] for red genes from (**a**). The 201 proteins shown formed networks of five or more proteins, with the number of interactions equal to the number of edges. (**d**) −Log_10_ FDR-corrected *p*-values (q-values) from one-sided Fisher’s exact tests analyzing gene set enrichment for each STRINGdb interaction cluster in (**c**). (**e**,**f**) AKR1C3 immunostaining in FFPE tissue adjacent to the sectioned region of case 1 (**e**) and non-neoplastic human brain (**f**). Image: 200×; scale bar: 50 μm. (**g**–**i**) Immunofluorescent co-staining of IDH1 R132H (white), AKR1C3 (green), and nuclei (blue (DAPI)) in case 1 demonstrating expression of AKR1C3 in malignant cells carrying the truncal IDH1 R132H mutation. Scale bar denotes 50 µm.

## 3. Discussion

We have described a novel strategy for deconstructing ITH through multiomic and multiscale analysis (MOMA) of serial tumor sections. By amplifying each tumor specimen into standardized biological replicates through serial sectioning, we obtained a large number of representative subsamples of each tumor with variable cellular compositions. Because section size and number can be tailored to experimental needs, MOMA provides flexibility for a variety of concurrent assays while preserving spatial information. We performed WES to identify mutations in a small number of distant sections, followed by deep sequencing of PCR amplicons spanning mutation sites to quantify SNV frequencies with a high level of confidence in a large number of sections. Although clusters of SNVs with highly correlated VAFs suggested distinct clones, integrative analysis of SNV and CNV frequencies (inferred from bulk DNA methylation data (case 1) or bulk RNA-seq data (case 2)) was required to accurately reconstruct clonal phylogenies. Using this approach, we identified the six most prevalent clonal populations of malignant cells in case 1 and five in case 2 and quantified their abundance in all tumor sections.

By comparing clonal abundance to genome-wide expression patterns over all tumor sections, we identified transcriptional profiles of distinct malignant clones in each case. Clone expression profiles were orthogonally validated through comparisons with normal human brains (case 1) and snRNA-seq using nuclei isolated from interpolated tumor sections (case 2). Analysis of these profiles revealed several interesting findings. First, markers of neural stem cells (radial glia) were most significantly enriched in the truncal clone. Second, gene sets representing transcriptional subtypes of glioblastoma [40] were significantly associated with distinct clones, suggesting stereotyped and potentially reversible malignant cell states that may reflect different microenvironments [55,56,57]. Third, single-nucleus expression signatures of mitosis were observed in most clones, suggesting they were not terminally differentiated. And fourth, genotyping nuclei analyzed by snRNA-seq revealed ependymal-like cells with oncogenic mutations, but no neuron-like cells with oncogenic mutations. To our knowledge, malignant ependymal cells have not previously been described in human astrocytomas. Because ependymal cells are normally derived from radial glia during early brain development [58,59], the presence of malignant ependymal cells (and the absence of malignant neurons) may point to a specific type of progenitor as the cell of origin for case 2. Alternatively, oncogenic mutations and/or resulting epigenetic changes may alter the differentiation potential of the cell of origin from its normal state. In either case, replication of our strategy for malignant glioma subtypes will clarify whether brain tumors with specific oncogenic mutations consistently produce malignant progeny with transcriptional profiles that resemble the same neurobiological cell types.

Although both cases were diagnosed as IDH-mutant grade 2 astrocytomas, they shared only one SNV (IDH1 R132H), which was truncal in both cases but lost from 21% of malignant cells (clone 3) in case 2 due to chr2q deletion. The extent of clonal heterogeneity, even for the same type of tumor, begs the question of how gene expression correlations to clonal abundance should be compared and integrated across cases. We reasoned that aggregating gene expression correlations to the truncal clone (equivalent to tumor purity) would identify the most specific and consistent transcriptional features of all malignant cells in both astrocytomas. This deceptively simple strategy has important implications for target discovery in cancer biology, because correlations between molecular abundance and tumor purity can be aggregated from huge numbers of bulk samples from similar cases that collectively represent many billions of cells. In statistical and economic terms, this strategy represents an efficient path for identifying robust molecular markers of malignant cells, including the non-oncogene dependencies that may vastly outnumber recurrently mutated genes [60].

We observed a highly significant genome-wide correlation between gene expression profiles of the truncal clone, suggesting that a core set of genes is consistently expressed by the founding population of malignant cells in IDH-mutant astrocytomas. This result is particularly striking given the biological and technical differences between case 1 (primary astrocytoma, microarray gene expression data) and case 2 (recurrent astrocytoma, RNA-seq gene expression data). Cross-referencing these genes with human PPI data [54] revealed distinct groups of interacting proteins that were significantly enriched with cancer-related pathways and processes, including WNT and MYC signaling, RNA splicing, and DNA repair. Furthermore, many of the genes whose expression patterns correlated most strongly with malignant cell abundance in both cases have been implicated in other types of cancer. For example, *AKR1C3*, which encodes a prostaglandin synthase involved in androgen production [61], is significantly upregulated and associated with poor outcomes in hepatocellular carcinoma [62], prostate cancer [63], and pediatric T-cell acute lymphoblastic leukemia [64]. These findings underscore the possibility that malignant cells from diverse cancers caused by distinct mutations may nevertheless share transcriptional dependencies that can be exploited therapeutically.

It is also important to note that transcriptional phenotypes of malignancy, including the upregulation of *AKR1C3*, persisted in clone 3 from case 2 despite loss of the driver mutation IDH1 R132H following chr2q deletion. IDH1 R132H perturbs genome-wide expression patterns by increasing production of the oncometabolite D-2-hydroxyglutarate [65], which competes with endogenous a-ketoglutarate to alter the activities of enzymes that are required to maintain normal DNA methylation [66]. Our findings support previous studies indicating that altered DNA methylation patterns can persist and perpetuate malignant phenotypes despite the loss of the mutated protein that caused them [42,43,44]. This example is illustrative because it highlights the limitations of current gene panels for cancer diagnostics, which provide binary calls for the presence or absence of common oncogenic mutations. In this case, such panels would indicate the presence of IDH1 R132H and recommend treatment that targets this mutation [67]. However, with more precise knowledge of this tumor’s evolutionary history, we can see that such treatment will fail for one-fifth of its malignant cells, since the mutated IDH1 protein is no longer there. In this case, it is these cells that will likely form the basis for therapeutic resistance.

There are several important methodological implications and limitations of our study. First, each tumor specimen analyzed in this study represents a fraction of the overall tumor volume; future efforts will analyze multiple, geographically distinct tumor subsamples to evaluate the consistency of clonal architecture. Second, MOMA requires many sections to detect meaningful correlations (e.g., 25 sections provide ~85% power to detect moderate correlations (|r| > 0.5, *p* < 0.05)) [68]. Third, DNA and RNA must be co-isolated from each section (i.e., from the same population of cells). Fourth, deep sequencing is required to establish high-confidence VAFs for SNVs, which are in turn required to estimate clonal frequencies. Fifth, limited variability in clonal frequencies may impact the ability to detect corresponding molecular signatures. Sixth, some types of mutations are not captured by our approach (e.g., noncoding SNVs, rearrangements, chromothripsis, etc.). And seventh, collinearity in the abundance of malignant and/or nonmalignant cell types may produce spurious correlations (which can be mitigated by differential coexpression analysis with normal tissue, as performed for case 1, or sectioning in multiple planes, as performed for case 2). For this reason, we also recommend validating transcriptional profiles of malignant clones using one or more orthogonal techniques. We found that multiscale integration of bulk sections and single nuclei allowed us to leverage the complementary strengths of each sampling strategy. Specifically, bulk sections facilitate multiomic integration while yielding robust molecular signatures driven by millions of cells, while single nuclei enable precise validation of predictions made from bulk data. However, the success of this approach depends on accurate classification of malignant nuclei. In our study, we found that popular algorithms for identifying cancer cells from inferred CNVs from gene expression data [25,49,50] were only ~60% accurate. Therefore, MOMA will benefit from improved algorithms for inferring malignancy and/or scalable methods for profiling gene expression and malignant cell genotypes in parallel.

## 4. Methods

### 4.1. Pseudobulk Analysis of scRNA-seq Data

Single-cell RNA-sequencing (scRNA-seq) data from Venteicher et al. [24] comprising 6243 cells from 10 IDH-mutant adult astrocytomas were downloaded from Gene Expression Omnibus (https://www.ncbi.nlm.nih.gov/geo/; accession ID = GSE89567, accessed on 27 March 2019). To generate a pseudobulk gene expression matrix from these data, 10% of all cells were randomly sampled and expression levels were summed for each gene from all sampled cells (this process was repeated 100× to generate a matrix with 100 pseudobulk samples). Using cell-class labels provided by the authors, the identities of all cells comprising each pseudobulk sample were tracked. Genome-wide differential expression analysis was performed by comparing all sampled malignant cells to all sampled nonmalignant cells using a two-sided t-test. In parallel, genome-wide gene coexpression analysis was performed as described [31]. Briefly, genome-wide biweight midcorrelations (bicor) were calculated using the WGCNA R package [69], and all genes were clustered using the flashClust [70] implementation of hierarchical clustering with complete linkage and 1—bicor as a distance measure. The resulting dendrogram was cut at a static height of 0.277, corresponding to the top 1% of bicor values. All clusters consisting of at least 10 genes were identified and summarized by their module eigengene [52] (i.e., the first principal component obtained by singular value decomposition) using the moduleEigengenes function of the WGCNA R package [69]. Highly similar modules were merged if the Pearson correlation of their module eigengenes was >0.85. This procedure was performed iteratively such that the pair of modules with the highest correlation > 0.85 was merged, followed by recalculation of all module eigengenes, followed by recalculation of all correlations, until no pairs of modules exceeded the threshold. The pseudobulk gene coexpression module most strongly associated with malignant cells was identified by maximizing the correlation between the module eigengene and the actual fraction of sampled malignant cells in each pseudobulk sample. Genome-wide Pearson correlations to this module eigengene (*k*_ME_ values) [52] were then calculated and compared to the results of single-cell differential expression analysis (t-values).

### 4.2. Sample Acquisition

The tumor specimen from case 1 (WHO grade II IDH-mutant primary astrocytoma) was obtained from a 40 y.o. female patient following surgical resection at the University of California, San Francisco (UCSF), along with the patient’s blood (UCSF case ID: SF9495). The tumor specimen from case 2 (WHO grade II recurrent astrocytoma, IDH-mutant) was obtained from a 58 y.o. male patient following surgical resection at UCSF, along with the patient’s blood (UCSF case ID: SF10711). Four postmortem control human brain samples from two brain regions (anterior cingulate cortex (ACC) and entorhinal cortex (EC)) were also obtained from routine autopsies of two individuals (41 and 75 y.o. females) at UCSF. Control samples were examined by a neuropathologist (E.J.H.) and found to exhibit no evidence of brain disease. Tissue samples for nucleic acid isolation were immediately frozen on dry ice without fixation. For tumor histology, a smaller subsample was formalin-fixed and paraffin-embedded (FFPE) using standard procedures. All tumor samples were obtained with donor consent in accordance with protocols approved on behalf of the UCSF Brain Tumor Center Tissue Core.

### 4.3. Serial Sectioning

Tissue cryosectioning was performed on a Leica CM3050S cryostat at −20 °C. Each sample was oversectioned to account for the possibility of low RNA quality or quantity from some cryosections; after excluding these (see below), most, but not all, analyzed sections were adjacent to one another. For the first case, 81 sections were cut and utilized as shown in Figure 2f. For each of the four control samples, ~120 sections were cut and 94 were utilized for gene expression profiling. For the second case, 140 sections were cut and utilized as shown in Figure 4f. In addition, the plane of sectioning for the second case was rotated 90 degrees at the halfway point to provide additional spatial variation (Figure 4f). These sectioning strategies resulted in 73% power to detect weak correlations (|r| > 0.3, *p* < 0.05) for case 1 and 83% power for case 2 [68]. To control for differences in the cross-sectional area of each tissue sample, section thickness was varied as needed to ensure sufficient and comparable amounts of nucleic acids could be extracted from sections for multiomic analysis. Quality control and usage information for all sections can be found in Appendix A (case 1), Appendix A (control samples), and Appendix A (case 2). Frozen sections were collected in RNase-free 1.7 mL tubes (Denville Scientific Inc, South Plainfield, NJ, USA) and stored at −80 °C.

### 4.4. Nucleic Acid Isolation and Quality Control

Tissue cryosections were thawed on ice and homogenized by pipette in QIAzol (Qiagen Inc., Valencia, CA, USA). For control samples, RNA was extracted from each section with the miRNeasy mini kit (Qiagen Inc., Valencia, CA, USA). For tumor samples, DNA and RNA were isolated simultaneously from each section with the AllPrep DNA/RNA/miRNA kit (Qiagen Inc., Valencia, CA, USA). All nucleic acid isolation from tissue sections was performed using a QIAcube automated sample preparation system according to the manufacturer’s instructions (Qiagen Inc., Valencia, CA, USA). Sections were processed in random batches of 12 on the QIAcube to avoid confounding section number with potential technical sources of variation associated with nucleic acid isolation.

Frozen blood was thawed and resuspended in red blood cell lysis solution (Qiagen Inc., Valencia, CA, USA). White blood cells were removed by centrifugation at 2000g for 5 min and repeated until white blood cells were depleted. Remaining red blood cells were resuspended in extraction buffer (50 mM Tris (pH of 8.0), 1 mM EDTA (pH of 8.0), 0.5% SDS, and 1 mg/ml Proteinase K (Roche, Nutley, NJ, USA)) and incubated overnight at 55 °C. The extracted DNA was treated with RNAse (40 μg/ml) (Roche, Nutley, NJ, USA) for 1 h at 37 °C before being extracted by phenol–chloroform and precipitated with ethanol. The resulting DNA was resuspended in TE buffer (10 mM 460 Tris, 1 mM EDTA (pH of 7.6)).

RNA and DNA were analyzed using a Nanodrop 1000 spectrophotometer (Thermo Scientific Inc., Waltham, MA, USA) to quantify concentrations, OD 260/280 ratios, and OD 260/230 ratios. Further validation of RNA and DNA concentrations was performed using the Qubit RNA HS kit and Qubit dsDNA HS kit on the Qubit 2.0 Fluorometer (Life Technologies Inc., Carlsbad, CA, USA). RNA integrity (RIN) was assessed using an Agilent 2100 Bioanalyzer (Agilent Technologies Inc., Santa Clara, CA, USA). Sections for which RIN ≥ 5 (case 1 median = 7.6, case 2 median = 8.3), OD 260/280 ratio ≥ 1.80 (case 1 median = 2.03, case 2 median = 1.94), and concentration by Nanodrop ≥ 9 ng/μL (case 1 median = 25.4 ng/μL, case 2 median = 9.25 ng/μL) were selected.

### 4.5. Whole-Exome Sequencing (WES) and Data Preprocessing

WES was performed at the UCSF Institute for Human Genetics Genomics Core facility (San Francisco, CA, USA). Exome libraries were prepared from 1 μg of genomic DNA from each analyzed section using the Nimblegen EZ Exome kit V3 (Roche, Nutley, NJ, USA). Paired-end 100 bp sequencing was performed on a HiSeq2500 sequencer (Illumina Inc., San Diego, CA, USA). The analysis of WES data was performed as previously described [71]. Briefly, paired-end sequences were aligned to the human genome (University of California, Santa Cruz, build hg19) using the Burrows-Wheeler Aligner (BWA, version 0.7.17) [72]. Uniquely aligned reads were further processed to achieve deduplication, base quality recalibration, and multiple sequence realignment with the Picard suite [73] and Broad Institute Genome Analysis ToolKit (GATK) [74]. After processing, a mean coverage of 131–151× and 104–122× was achieved for case 1 and case 2, respectively.

### 4.6. Single-Nucleotide Variant (SNV) and Small Insertion/Deletion (Indel) Calling Workflow

SNVs were identified using MuTect (version 1.1.5) [75], and indels were identified with Pindel [76] using default settings. SNVs were further filtered to only retain variants with frequency > 0.10 in at least one tumor section and <6 variant reads in the patient’s blood. Indels were filtered to only retain variants with >5 variant reads in a given tumor section and <13 total reads in the patient’s blood. If multiple indels were detected at the same genomic location, only the indel with the most supporting reads was retained. Identified mutations were annotated for their mutational context using ANNOVAR (version 2.4) [77] and were also cross-referenced with dbSNP [78] (build ID: 132) and the 1000 Genomes project [79] (Phase 1). SNV and indel events were converted to hg38 coordinates and assigned HGVS-compliant names using Ensembl’s Variant Effect Predictor (version 104.3) [80].

### 4.7. Droplet Digital PCR (ddPCR) 

Variant allele frequencies (VAFs) of the IDH1 R132H mutation were determined in 69 tumor sections from case 1 and the patient’s blood using the PrimePCR IDH1 R132H mutant assay and the QX100 Droplet Digital PCR system (Bio-Rad Inc., Hercules, CA, USA). An initial serial dilution of a positive control was performed to optimize the input concentration of genomic DNA from each section and to assess the reliability of the assay. Duplicate reactions were performed to quantify the reproducibility of the assay (Appendix A). Data were analyzed, and 95% Poisson confidence intervals were calculated using QuantaSoft software (version 1.7) (Bio-Rad Inc., Hercules, CA, USA).

### 4.8. Amplicon Sequencing (amp-seq) and Data Preprocessing

Groups of mutations with similar allele frequency distributions in WES data were identified by hierarchical clustering. Biweight mid-correlations (bicor) were used to estimate the proximities of somatic mutations, and 1-bicor was used as a dissimilarity measure. A subset of representative mutations from distinct clusters was validated by Sanger sequencing and deep sequencing of PCR amplicons (amp-seq) derived from tumor sections and the patient’s blood. Primers were designed using Primer-BLAST [81] to yield an amplicon of around 500 bp (case 1) or 100 bp (case 2) with the mutation located within the center of the amplicon (Appendix A). Amplicons were generated for 42 mutations in case 1 (n = 69 sections; Appendix A) and 75 mutations in case 2 (n = 85 sections; Appendix A). For case 1, the mutation-containing region was amplified by PCR using the FastStart high-fidelity PCR system (Roche, Nutley, NJ, USA) or the GC-Rich PCR system (Roche, Nutley, NJ, USA) as instructed by the manufacturer using specific annealing temperatures (Appendix A). The resulting amplicons were purified using the NucleoSpin gel and PCR cleanup kit following the manufacturer’s instructions (Macherey-Nagel Inc., Bethlehem, PA, USA) and submitted for Sanger sequencing with the same primers used to generate the amplicons. For case 2, 50ng of gDNA was used as template per sample in each reaction, and 35 cycles of PCR amplification were performed with KAPA HiFi HotStart Ready Mix (2x, KAPA Biosystems, Wilmington, MA, USA). Multiplexed PCR reactions were purified using a 2X volume ratio of KAPA pure SPRI beads (KAPA Biosystems, Wilmington, MA, USA). Purified PCR reactions were quantified using the Qubit dsDNA HS kit and Qubit 2.0 fluorometer. For both cases, the concentration of each amplicon was adjusted to 0.2 ng/μL. Barcoded libraries for each section were generated using the Nextera XT DNA Kit (Illumina Inc., San Diego, CA, USA). After library preparation, the barcoded libraries were pooled using bead-based normalization supplied with the Nextera XT kit. The pooled libraries were sequenced with paired-end 250 bp reads in a single flow cell on an Illumina MiSeq (Illumina Inc., San Diego, CA, USA) in case 1 and an Illumina HiSeq 4000 in case 2. In case 1, libraries were sequenced in two runs, whereas all amplicons were sequenced in the same run for case 2. Sequence reads were demultiplexed and basecalled using “bcl2fastq” (Illumina Inc., San Diego, CA, USA). FASTQ files were aligned to a custom genome (based on the amplicon sequences) using BWA-MEM [82]. The SAMtools suite [83] was used to create and index BAM files and create pileup files based on reads with a base quality score > 30. Read counts supporting the reference or variant within each amplicon were determined using the read counts function from VarScan 2 [84] and these counts were used to calculate VAFs.

### 4.9. Downsampling Analysis of amp-seq Data

Amplicon reads originating from the reference or alternative alleles for *IDH1* or *TP53* were randomly downsampled to various coverage levels (n = 1000 random downsamples per coverage level) for each section to quantify the effect of reduced coverage on VAF estimates. VAFs were recalculated for each downsampled coverage level and compared to full coverage VAF estimates over all sections using Pearson’s correlation or root mean square error (RMSE), as illustrated in Appendix A (case 1) and Appendix A (case 2).

### 4.10. Hierarchical Clustering of Variant Allele Frequencies (VAFs)

Groups of mutations with similar VAF patterns were identified by hierarchical clustering over all tumor sections. VAFs were clustered with Ward’s D method and 1—Pearson’s correlation as a dissimilarity measure. The number of clusters was determined from the consensus of elbow [85] and silhouette plot [86] methods, using the cluster package in R [87].

### 4.11. DNA Methylation Data Production and Preprocessing

The sample order of genomic DNA from serial sections of case 1 was randomized to avoid confounding section number with potential sources of technical variation. DNA was concentrated with Genomic DNA Clean & Concentrator 10TM columns (Zymo Research, Irvine, CA, USA) in batches of 12 samples, resulting in approximately two-fold concentration (median concentration after processing: 45 ng/μL). The sample order was randomized again, and concentrated DNA was shipped on dry ice to the University of California, Los Angeles (UCLA) Neurogenomics Core facility (Los Angeles, CA, USA) for analysis using Illumina 450K microarrays (Illumina Inc., San Diego, CA, USA). 

Raw idat files were processed using the ChAMP R package [88]. Initial probe filtering was performed using the load.champ R function [89,90,91]. Probes with detection *p*-value > 0.01 (11,799 probes) or beadcount < 3 in at least 5% of samples were removed (n = 760), leaving 461,797 probes for analysis. The Illumina 450K microarrays contain two different assay types (Infinium I and Infinium II). Each assay has different sensitivity and dynamic range, which means that joint normalization leads to type II bias due to the lower sensitivity of the Infinium II assay [92]. We therefore performed beta-mixture quantile normalization (BMIQ) using the “champ.norm” function from ChAMP, which accounts for the different assay types [93]. 

Additional preprocessing of the methylation data was performed with the SampleNetwork R function [94], which identifies outlying samples, performs data normalization, and corrects for technical batch effects. The standardized sample network connectivity (Z.K) criterion was used to exclude one outlying sample (section #69, whose DNA concentration was substantially lower than that of other sections), leaving 68 sections. No batch effects associated with ArrayID or ArrayPosition were observed.

### 4.12. Gene Expression Data Production and Preprocessing

Total RNA from case 1 (n = 69 sections) was shipped on dry ice to the UCLA Neurogenomics Core facility (Los Angeles, CA, USA) for analysis using Illumina HT-12 v4 human microarrays (Illumina Inc., San Diego, CA, USA). The order of the sections was randomized prior to shipment to avoid confounding potential technical artifacts with potential biological gradients of gene expression. Two control samples from the same pool of total human brain RNA (Ambion FirstChoice human brain reference RNA Cat#AM6050, Life Technologies Inc., Carlsbad, CA, USA) were included with each of the five datasets. For each of the five datasets (case 1 and four control samples), all microarray samples (n = 72–96/dataset) were processed in the same batch for amplification, labeling, and hybridization. Amplification was performed using the Ambion TotalPrep RNA amplification kit (Life Technologies Inc., Carlsbad, CA, USA). Raw bead-level data were minimally processed by the UCLA Neurogenomics Core facility (no normalization or background correction) using BeadStudio software (version 3.2) (Illumina Inc., San Diego, CA, USA).

For each dataset, the minimally processed expression data were further preprocessed using the SampleNetwork R function [94]. Using the standardized sample network connectivity (Z.K) criterion [94], the following numbers of outliers were removed from each dataset: ACC1 (n = 2), ACC2 (n = 11), EC1 (n = 0), EC2 (n = 2), and case 1 (n = 1). Exclusion of outliers resulted in the following numbers of remaining sections in each dataset: ACC1 (n = 92), ACC2 (n = 83), EC1 (n = 94), EC2 (n = 92), and case 1 (n = 69). After removing outliers, each dataset was quantile normalized [95], and technical batch effects were assessed [94]. Significant batch effects (*p* < 0.05 after Bonferroni correction for univariate ANOVA) were corrected using the ComBat R function [96] with no covariates as follows: ACC1 = ArrayID, ACC2 = ArrayID, EC1 = ArrayID and ArrayPosition, and EC2 = QCBatch and ArrayID. No batch effects were observed for case 1. Multiple technical batch effects were corrected sequentially. Analysis was restricted to 30,425 probes that were re-annotated [97] as having either “perfect” (n = 29,272) or “good” (up to two mismatches; n = 1153) sequence alignment to their target transcripts. Probes were further collapsed to unique genes (n = 20,019) by retaining one probe per gene with the highest mean expression for all sections.

For case 2, RNA-sequencing was used to profile gene expression for all sections (n = 96). Full-length RNA was made into libraries using the KAPA stranded mRNA library prep kit (Roche, Nutley, NJ, USA) following the manufacturer’s instructions, with a mean insert size of 300 bp. One ng of library (composed of library and ERCC spike-in controls, Life Technologies Inc., Carlsbad, CA, USA) was added as input, and all libraries were normalized according to the manufacturer’s instructions. During this process, samples were randomized in both section order and plane to avoid conflating biological and technical covariates. Sequencing was performed on eight lanes of a HiSeq4000 at the Center for Advanced Technology (CAT) at UCSF with single-end 50 bp sequencing using dual-index barcoding.

Reads were assessed with FastQC to ensure the quality of sequencing data by verifying high base quality scores, lack of GC bias, narrow distribution of sequencing lengths, and low levels of sequence duplication or adapter sequences [98]. Next, reads were subjected to adapter trimming using Cutadapt [99] with minimum length = 20 and a quality cutoff of 20. Reads were subsequently aligned using default settings with the Bowtie2 program [100] to the Genome Reference Consortium Human Build 37 [101]. Finally, an expression matrix was generated using the FeatureCounts program with UCSC’s library of genomic features [102] (n = 23,900 features). Genes with zero variance were removed (n = 30). Data were normalized with the RUVg package, regressing out 10 factors derived from principal component analysis of the ERCC spike-in control expression matrix [103]. The number of factors was determined empirically by evaluating relative log-expression (RLE) plots and gene–gene correlation distributions. Finally, the SampleNetwork R function [94] was used to identify and remove six outlier sections based on the standardized sample connectivity criterion (Z.K).

### 4.13. Copy Number Analysis by qPCR

The copy numbers for *TP53* and *ACCS* in case 1 were determined by SYBR Green-based qPCR. Primers were designed using Primer-BLAST [81] and positioned immediately adjacent to but not including the SNV (ACCS F: TCTCTATGGCAACATCCGGC, R: CAGCCATGCAGCAACAGAAG; RPPH1 F: CGGAGGGAAGCTCATCAGTG, R: CCGTTCTCTGGGAACTCACC, TERT F: CTCGGATCATGCTGAGGACC, R: TTGTGCAATTCTGTGCCAGC, TP53 F: CAGTCACAGCACATGACGGA, R: GGGCCAGACCTAAGAGCAAT). qPCR was performed on genomic DNA from all 69 tumor sections and the patient’s blood using the LightCycler 480 SYBR Green I master mix and LightCycler 480 qPCR machine according to the manufacturer’s recommendations (Roche, Nutley, NJ, USA). Measurements were taken in triplicate, and data were analyzed using the standard curve method. Copy numbers were determined for *TP53* and *ACCS* and two control genes on different chromosomes: ribonuclease P RNA component H1 (*RPPH1*) and telomerase reverse transcriptase (*TERT*). Relative copy number was determined by dividing the mean copy number of *TP53* and *ACCS* by the mean copy number of each reference gene separately to obtain a ratio and multiplying the ratio by two to obtain the diploid chromosome number. The relative copy number normalized to one of the reference genes (*RPPH1*) is shown in Appendix A.

### 4.14. Copy Number Variation (CNV) Calling (Bulk Data)

CNVs were quantified using multiple technologies and algorithms to generate reliable estimates. Although WES remains the gold-standard method for calling CNVs, DNA methylation and RNA-seq data provide cost-effective options that can be triangulated with sparse WES data to reduce false positives. Unless otherwise noted, default parameters were used. For case 1 we used the champ.CNA function, included with the ChAMP R package [39], to call CNVs from DNA methylation data. For both cases, we called CNVs from exome data using FACETS [38] with critical values of 25 (case 1) and 450 (case 2). Finally, we used CNVkit with circular binary segmentation to call CNVs from bulk RNA-seq data [45,104,105].

### 4.15. Generation of Clonal Trees with Corresponding Frequencies

CNVs were filtered to ensure that they were called in exome data and either DNA methylation data (case 1) or RNA-seq data (case 2) and covered more than 10% of a chromosomal arm. CNV coordinates were defined based on the intersection of ranges from both methods (Appendix A). Using the frequencies of CNV/SNV mutations and tumor purity estimated from the *TP53* locus as input to PyClone [35], we determined cluster membership for SNP and CNV events. We then used the PyClone output as the input to the CITUP algorithm [36] to generate the most likely clonal tree (i.e., the tree with the minimum objective value) and derive clonal frequencies. In cases in which there was an approximate tie between objective values, the tree was manually chosen based on biologically plausible principles. To visualize results, we used the data.tree [106] and DiagrammeR [107] packages in R.

### 4.16. Gene Coexpression Network Analysis

Genome-wide biweight midcorrelations (bicor) were calculated using the WGCNA R package [69] for case 1 (n = 20,019 genes) and case 2 (n = 23,870 genes). All genes were clustered using the flashClust (version 1.01-2) [70] implementation of hierarchical clustering with complete linkage and 1—bicor as a distance measure. Each resulting dendrogram was cut at a static height (0.875 for case 1 and 0.562 for case 2) corresponding to the top 30% and 20% of values of the correlation matrix for case 1 and case 2, respectively. All clusters consisting of at least 15 members for case 1 or five members for case 2 were identified and summarized by their module eigengene [52] (i.e., the first principal component obtained by singular value decomposition) using the moduleEigengenes function of the WGCNA R package [69]. Highly similar modules were merged if the Pearson correlation of their module eigengenes was > 0.80. This procedure was performed iteratively such that the pair of modules with the highest correlation > 0.80 was merged, followed by recalculation of all module eigengenes, followed by recalculation of all correlations, until no pairs of modules exceeded the threshold (case 1: Appendix A; case 2: Appendix A).

### 4.17. Module Enrichment Analysis

The WGCNA measure of module membership, *k*_ME_, was calculated for all genes with respect to each module. *k*_ME_ is defined as the Pearson correlation between the expression pattern of a gene and a module eigengene and therefore quantifies the extent to which a gene conforms to the characteristic expression pattern of a module [52] (case 1: Appendix A; case 2: Appendix A). For enrichment analyses, module definitions were expanded to include all genes with significant *k*_ME_ values, with significance adjusted for multiple comparisons by correcting for the false-discovery rate [108]. If a gene was significantly correlated with more than one module, it was assigned to the module for which it had the highest *k*_ME_ value. Enrichment analysis was performed for all modules using a one-sided Fisher’s exact test as implemented by the fisher.test R function.

### 4.18. Lasso Modeling of Gene Expression

The machine learning variable selection method lasso (least absolute shrinkage and selection operator) and group lasso were performed using the R package Seagull [109,110,111]. Modeling was performed for each case with gene expression patterns as dependent variables and clonal frequency vectors as independent variables. For case 1, clone 2 was excluded from modeling due to its low frequency and clone 6 was excluded since it was defined by a single CNV. Because clone 1 corresponds to the tumor purity vector, which represents the major vector of variation in this dataset, many genes experience inflated correlations to clone 1. To counteract this effect, group lasso was performed. The truncal clone (clone 1) was placed in its own group, and all remaining clones to be modeled were placed in a separate group. This procedure improved modeling performance for case 1 (Appendix A) but not case 2 (Appendix A), which may reflect the greater variance in tumor purity for case 1. As such, modeling results for case 2 presented in the manuscript are derived from the regular lasso model. For each gene, models were bootstrapped (n = 100) to address collinearity among clonal frequency vectors [112] (as shown in Appendix A). We also generated empirical null distributions for model performance by permuting each gene’s expression profile prior to bootstrapping (n = 100).

When performing group lasso modeling, only models with one surviving clonal frequency vector (not including the truncal clone) were considered. When performing lasso modeling, only models with one surviving clonal frequency vector were considered. To quantify model stability, we calculated the number of times out of 100 bootstraps that the most frequent surviving independent variable was the sole surviving variable. This stability metric was calculated for all gene models, including the permuted models. From the resulting distributions of stability values, a 5% FDR threshold was determined. For case 1, the stability value of 73 represents the point beyond which 5% or less of the models were permuted models. Similarly, for case 2, the 5% FDR threshold for the stability metric was 45. Gene set enrichment analysis was performed via a one-sided Fisher’s exact test for all genes with significant model stability for the same clonal frequency vector (Appendix A for case 1 and case 2, respectively), with genes separated by the sign of the coefficient for the independent variable (Appendix A for case 1 and case 2, respectively).

### 4.19. Differential Gene Coexpression Analysis

Using the WGCNA R package [69], pairwise biweight midcorrelations (bicor) were calculated among all 30,425 high-quality probes over all sections (n = 69–94) in each of five datasets (case 1 + four normal human brain samples), generating five identically proportioned correlation matrices (30,425 × 30,425). These correlations were then scaled to lie between [0, 1] using the strategy of Mason et al. [113]. To identify gene coexpression relationships that were present in tumor but absent or weaker in normal human brains, each scaled bicor matrix produced from normal human brains was subtracted [114] from the scaled bicor matrix produced from case 1, resulting in four “subtraction matrices”, or SubMats. The consensus of the four SubMats was formed by taking the minimum value at each point in the four matrices using the parallel minimum (pmin) R function, and the resulting “Consensus SubMat” was used as input for gene coexpression analysis (Appendix A). By definition, gene coexpression modules identified with this strategy consist of groups of genes with expression patterns that are highly correlated in the astrocytoma but not in any of the normal human brain samples (Appendix A).

Probes in the Consensus SubMat were clustered using the flashClust [70] implementation of a hierarchical clustering procedure with complete linkage and 1—Consensus SubMat as a distance measure. The resulting dendrogram was cut at a static height of ~0.38, corresponding to the top 2% of values in the Consensus SubMat. All clusters consisting of at least 10 members were identified and summarized by their module eigengene [52] using the moduleEigengenes function of the WGCNA R package (version 1.70-3) [69]. Highly similar modules were merged if the Pearson correlation of their module eigengenes was >0.85. This procedure was performed iteratively such that the pair of modules with the highest correlation > 0.85 was merged, followed by recalculation of all module eigengenes, followed by recalculation of all correlations, until no pairs of modules exceeded the threshold. The WGCNA [69] measure of intramodular connectivity (*k*_ME_) was calculated for all probes (n = 47,202) with respect to each module by correlating each probe’s expression pattern across all 69 tumor sections with each module eigengene.

### 4.20. Single-Nucleus DNA-Sequencing and Analysis

Three sections from case 2 (sections 29 and 113/115, which were combined) were analyzed using the MissionBio, Inc. (MissionBio, San Francisco, CA, USA) Tapestri microfluidics platform for single-nucleus DNA amplicon sequencing [46]. Using an in-house protocol, 4433 (section 29) and 3736 (sections 113/115) nuclei were extracted and recovered for analysis with the Mission Bio AML panel, which includes primers flanking one *IDH1* and two *TP53* loci. In addition, chr17 chromosomal copy number changes and *TP53* zygosity were inferred from a germline heterozygous intronic mutation upstream of TP53 G245V that happened to fall within the targeting panel (NC_000017.11:g.7674797T>A). Sequencing was performed on an Illumina MiSeq (Illumina Inc., San Diego, CA, USA), yielding an average of 6801 (section 29) or 6433 (sections 113/115) reads per nucleus, with alignment rates of ~90%. Hierarchical clustering of nuclei for mutations of interest was performed separately for section 29 and sections 113/115 using complete linkage and Euclidean distance, with *k* = 4 chosen based on silhouette [86] and elbow plots [85]. Genotype calls for the clusters were manually annotated as described in Appendix A and Appendix A.

### 4.21. Single-Nucleus RNA-Sequencing and Analysis

#### 4.21.1. Library Prep and Sequencing

Four sections (17, 53, 93, 117) from case 2 were used to generate single-nucleus RNA-seq (snRNA-seq) data. Our approach was adapted from TARGET-Seq [47], a protocol utilizing dual-indexing of sample barcodes and unique molecular identifiers (UMIs) of captured transcripts. Briefly, for each section, lysis was performed by Dounce homogenization with staining of nuclei by Hoescht3342 and subsequent flow-sorting into three 96-well plates per section. Each plate was randomized and subsequently processed individually and in random order. We used the SmartScribe kit (Takara Bio USA, San Diego, CA, USA) for RT-PCR, followed by PCR with the SeqAmp PCR kit (Takara Bio USA, San Diego, CA, USA). Unlike TARGET-Seq, the RT reaction was performed using only polyA primers (Appendix A). ERCC spike-in control RNA was added to the wells according to manufacturer’s instructions to facilitate identification and correction of batch effects. Wells for each plate were pooled in equivolume proportions, and an Agilent 2100 Bioanalyzer (Agilent Technologies Inc., Santa Clara, CA, USA) was used to assess sample quality. cDNA concentrations were quantified using a Qubit 2.0 Fluorometer with the dsDNA-High Sensitivity kit (Life Technologies Inc., Carlsbad, CA, USA), yielding mean cDNA concentration of 1ng/ul. Concentrations were normalized prior to tagmentation (Nextera Kit, Illumina Inc., San Diego, CA, USA) and amplification of 3′ ends, as in TARGET-Seq [47]. Sequencing was performed using the 150-cycle high-throughput kit on an Illumina NextSeq550 at SeqMatic (Fremont, CA, USA) with dual-indexed sequencing and read parameters as in TARGET-seq.

#### 4.21.2. Data Preprocessing

snRNA-seq raw reads were demultiplexed and basecalled using “bcl2fastq” (Illumina Inc., San Diego, CA, USA). Barcodes were filtered using the “umi_tools” package [115] whitelist function, with a Hamming distance of 2 and the density knee method to determine the number of true barcodes. A total of 809/1152 nuclei (70.2%) passed this initial quality control step. Reads were assessed with FastQC to ensure the quality of sequencing data by verifying high base-quality scores, lack of GC bias, narrow distribution of sequencing lengths, and low levels of sequence duplication or adapter sequences [98]. Next, reads were subjected to adapter trimming using the Trimmomatic algorithm [116] with a minimum length of 30 and a minimum quality of 4 with a 15 bp sliding window; otherwise, the default settings were used. A mean of 445,082 reads/nucleus was achieved at this stage. Reads were subsequently aligned using ENCODE RNA-seq settings (except for outFilterScoreMinOverLread, which was set to 0) with the STAR program [117] to the Genome Reference Consortium Human Build 38 [101]. Finally, an expression count matrix was generated using the FeatureCounts program [118] with Gencode’s library of gene features (version 21) [119] subset using the “gene” attribute (n = 60,708 features). Deduplication of UMIs was performed using a custom R script, resulting in a mean number of 206,638 unique reads/nucleus, with a 46% deduplication rate. Features with counts less than one in more than 90% of cells were removed (n = 57,021 final features). Data were normalized with the RUVg package, regressing out 10 factors derived from PCA of the ERCC spike-in control expression matrix [103]. Normalized counts were further processed using the Sanity package (version 2.0) [27], with 1000 bins and a minimum and maximum variance of 0.001 and 1000, respectively. Internuclear distance was determined using the Sanity_distance function with a signal-to-noise parameter of 1 and inclusion of error bars.

#### 4.21.3. snRNA-seq Clustering and Differential Expression Analysis

snRNA-seq data were hierarchically clustered using the hclust function in R with Ward’s method and the distance metric derived by Sanity [27]. This distance metric uses a Bayesian approach by giving less weight to gene expression estimates with large error bars when calculating cell distances. The optimal number of clusters (*k* = 12) was determined using elbow [85] and silhouette plots [86] with the cluster package in R [87]. Differential expression analysis (t.test) was performed between each cluster and all other clusters using Sanity-adjusted expression values for all genes. The resulting distributions of t-values were then compared for genes comprising the bulk coexpression modules most strongly associated with each malignant clone/nonmalignant cell type and all other genes (white and black distributions, respectively, in Appendix A; significance was evaluated with a one-sided Wilcoxon rank-sum test). Module genes were defined as those that were significantly correlated with corresponding bulk coexpression module eigengenes as determined by the FDR threshold [108]. If a gene was significantly correlated with more than one module eigengene, it was assigned to the module for which it had the highest *k*_ME_ value.

#### 4.21.4. CNV Calling

The snRNA-seq count matrix was used as input to CopyKat [49]. Nuclei snRNA-seq clusters determined to be nonmalignant by snAmp-seq were used as normal control cells. “KS.cut” was set to 2, “ngene.chr” was set to 20, and Ensembl gene names were used. InferCNV [25] was provided with a vector of nonmalignant cells (as previously determined) based on clustering and snAmp-seq in “subclusters” mode, with a cutoff parameter of 1, denoising turned on, “ward.D” as clustering method, “qnorm” as subcluster partition method, and tumor subcluster *p*-value of 0.05. The Hidden Markov model was not used. The program CaSpER [50] was run with the raw snRNA-seq count matrix as input and default settings, again using snRNA-seq clusters of nuclei determined to be malignant by snAmp-seq as negative controls. For each of these algorithms, the outputs were clustered based on Euclidean distance using Ward’s D method. Clusters with no CNV signal were labeled nonmalignant while all other clusters were presumed to represent malignant cells. 

Sensitivity and specificity were calculated using the snAmp-seq data as ground truth. True positives (TP) were defined as the intersection of malignant calls by the CNV calling algorithm and the snAmp-seq data. True negatives (TN) were defined as the intersection of nonmalignant calls by the CNV calling algorithm and the snAmp-seq data. False negatives (FN) and false positives (FP) were similarly defined. Nuclei with insufficient data were excluded from the analysis. Sensitivity was defined as follows: TP/(TP + FN), while specificity was defined as follows: TN/(TN + FP). Accuracy was defined as (TP + TN)/(TP + FP + TN + FN).

#### 4.21.5. UMAP and Trajectory Analysis

UMAP was performed for all nuclei (n = 809) with a starting seed of 15, 30 neighbors, a spread of 3, a minimum distance of 2, and 1—Pearson correlation as a distance metric using the “uwot” R package (version 0.1.16) [120] after selecting the first 30 principal components of the Sanity-corrected expression matrix including all genes. UMAP was also performed separately for all cells associated with malignant clusters using the Sanity-corrected expression matrix. After selecting the first 15 principal components, the “uwot” package was used with a seed of 15, 20 neighbors, a spread of 3, a minimum distance of 2, and 1—Pearson correlation as the similarity metric. All other settings were left as defaults. Trajectory analysis was performed with the Slingshot R package [51] on the UMAP plot. The “simple” distance method was used, and all other parameters were left as their default values.

#### 4.21.6. Gene Set Enrichment Analysis

Enrichment analysis (one-sided Fisher’s exact test) was performed for each snRNA-seq cluster using genes that were differentially expressed in that cluster relative to all other clusters using a one-sided Wilcoxon rank-sum test. Resultant *p*-values were further FDR-corrected to q-values [108]. Gene sets used for enrichment analysis are listed in Appendix A.

#### 4.21.7. Amp-Seq Genotyping

Single-nucleus amplicon-seq (snAmp-seq) of single-nucleus cDNA was adapted from the TARGET-seq protocol [47]. Primers flanking the following mutations (marking the truncal clone) were designed with Primer3 [121]: IDH1 R132H, TP53 G245V, and RUFY1 K218N (Appendix A). To overcome lack of heterogeneity in sequencing, random spacers were added to the beginning (5′ end) with 0–5 nucleotides from the sequence CGTAC. Finally, a common sequence was added to the 5′ end of the primer for a second round of PCR (Appendix A). We selected wells that passed QC for snRNA-seq analysis and processed each plate separately and in random order. Amplification of the first round of PCR was performed with the KAPA 2G Ready Mix (Roche Inc., Nutley, NJ, USA) with the same PCR program as for TARGET-Seq [47]. The program “Barcrawl” [122] was used to create custom dual-index barcodes for the amplification PCR. At this stage, 10% of wells were checked using an Agilent 2100 Bioanalyzer (Agilent Inc., Santa Clara, California) to determine whether products of appropriate size were produced. All wells were quantified with a Qubit 2.0 Fluorometer using the dsDNA-High Sensitivity kit (Life Technologies Inc., Carlsbad, CA, USA) and normalized prior to the next step. The second round of PCR used custom sequencing primers that were partially complementary to the previous sequences, with custom dual-index barcodes generated from BarCrawl [122] and Illumina P5/P7 sequences. Sequencing was performed using a 300 cycle MiSeq v2 Nano kit on an Illumina MiSeq (Illumina Inc., San Diego, CA, USA).

snAmp-seq data were demultiplexed and basecalled using “bcl2fastq” (Illumina Inc., San HDiego, CA, USA). Reads were assessed with FastQC to ensure the quality of sequencing data by verifying high base quality scores, lack of GC bias, narrow distribution of sequencing lengths, and low levels of sequence duplication or adapter sequences [98]. Next, reads were subjected to adapter trimming using the Trimmomatic algorithm [116] with a minimum length of 30 and a minimum quality of 4 with a 15 bp sliding window; otherwise, default settings were used [116]. Reads were subsequently aligned with the STAR program to a custom version of the genome containing only the amplicons of interest. Default parameters were altered such that no multiple alignments or splicing events were allowed. The median number of reads per nucleus for each amplicon was as follows: IDH1 R132H: 177; TP53 G245V: 246; and RUFY1 K218N: 209. Read counts supporting the reference or variant allele within each amplicon were determined using the read counts function from VarScan 2 [84], and these counts were used to calculate variant frequencies. Nuclei were sorted into three categories: called nuclei (calls by VarScan 2 of two or more mutant or two or more wild-type (WT) calls of the three loci with either one or zero indeterminate calls), discrepant nuclei (two WT and one mutant call), and insufficient data nuclei (two or more loci in which VarScan 2 was unable to call a genotype). The breakdown for these categories is as follows: 75% were called nuclei, 1% were discrepant nuclei, and 24% were insufficient data nuclei (Appendix A).

### 4.22. Inter-Case Analysis

Combined Pearson correlations to tumor purity for the 15,288 genes shared between case 1 and case 2 were determined by calculating the weighted average of the z-scores produced by Fisher’s transformation, dividing this value by the joint standard error, and applying the inverse Fisher transformation [31]. To define significant genes for enrichment analysis (Figure 9a,b), a minimum absolute value for Pearson’s correlation of >0.3 or <−0.3 was required in both cases along with an FDR-corrected q-value of <0.05. Enrichment analysis was performed as described above, with gene sets listed in Appendix A. Significant positively correlated genes were subjected to protein–protein interaction (PPI) analysis using the STRING database [53]. We used the STRINGdb [53,54], network [123], intergraph [124], and ggnetwork [125] packages to visualize the results of STRING PPI analysis. The “physical” network flavor and minimum score of 900 were utilized to guarantee that all depicted interactions were actual PPIs with experimental evidence. Clusters with more than five members were chosen from the set of interaction clusters generated from all genes that had positive correlations and passed the correlation cutoffs listed above. Enrichment analysis of PPI clusters was performed as described above, with gene sets listed in Appendix A.

### 4.23. Histology and Immunostaining

Tumor tissue was fixed in 10% neutral-buffered formalin, processed, and embedded in paraffin. Tumor sections (5 μm) were prepared and stored at −20 °C prior to use. Hematoxylin and eosin staining was performed using standard methods. As part of clinical evaluation, the proliferative index and TP53 mutation status were estimated based on review of immunostained slides for KI67 or TP53, respectively. Briefly, in regions with increased signal, the percent of tumor cell staining was estimated based on review of ten 200× fields.

Anti-AKR1C3 was selected based on statistical considerations pursuant to bioinformatic analyses and after preliminary validation of efficacy in human tissue via The Human Protein Atlas [126] (http://www.proteinatlas.org; accessed on 3 September 2019). Primary antibodies and conditions were IDH1 R132H (DIA-H09, Dianova, mouse clone H09, dilution 1:50); AKR1C3 (Catalog# AB84327, Abcam, rabbit polyclonal, dilution 1:600 for single immunohistochemistry and 1:1200 for dual immunofluorescence); and TP53 (1:25, Novocastra, catalog # P53-D07-L-CE-H). Heat antigen retrieval was performed in Tris-EDTA at pH of 8. Following antigen retrieval, sections for immunohistochemistry were treated with 3% methanol-hydrogen peroxide at 22 °C for 16 min. 

All immunostaining and multiplex immunostaining were performed using a Discovery XT autostainer or Benchmark XT (Ventana Medical Systems, Inc., Oro Valley, AZ, USA). For signal detection, the Multimer HRP kit (Ventana Medical Systems, Inc., USA) followed by either DAB or fluorescent detection kits. Fluorophores with the least autofluorescence on FFPE tissue were selected to minimize false positives: Cyanine 5 (Cy5) (DISCOVERY CY5 Kit, Cat#760238, Roche Diagnostics Corporation, Indianapolis, USA) and rhodamine (DISCOVERY Rhodamine Kit, Cat#760233, Roche Diagnostics Corporation, Indianapolis, IN, USA). Slides were then counterstained with DAPI (Sigma Aldrich, St. Louis, MI, USA) at 5 μg/mL in PBS (Sigma Aldrich, USA) for 15 min, mounted with prolong Gold antifade mounting media reagent (Invitrogen, Waltham, MA, USA) and stored at −20 °C prior to imaging. Positive and negative controls were included for each marker. Images of stained slides were acquired using either a light microscope (Olympus BX41 microscope using UC90 Cooled CCD 9 Megapixel camera) or Zeiss Cell Observer epifluorescence microscope equipped with an AxioCam 506M camera and an Excellitas X-Cite 120Q light source and processed with Photoshop CS6 (Adobe systems, San Jose, CA, USA). Nonmalignant tissue analyzed in Figure 9 was obtained from a patient with epilepsy and corresponds to normal tissue adjacent to epileptic foci.

### 4.24. Data Analysis and Figure Production

Unless otherwise stated, all analyses were performed in the R computing environment (https://www.r-project.org, version 4.1.2). Figures were produced with the aid of the R packages ggplot2 [127], data.table [128], RColorBrewer [129], gridExtra [130], ComplexHeatmap [131], Circlize [132], and ggsignif [133].

## 5. Conclusions

We have described a novel, flexible, and statistically motivated strategy for deconstructing intratumoral heterogeneity through multiomic and multiscale analysis (MOMA) of serial tumor sections. MOMA uses deep sampling, deep sequencing, and integrative analysis to reconstruct and internally cross-validate the phylogenies, spatial distributions, and core molecular properties of distinct malignant clones. By genotyping nuclei analyzed by snRNA-seq for truncal mutations, we show that commonly used algorithms for identifying cancer cells from single-cell transcriptomes may be inaccurate. We propose that aggregating genome-wide expression correlations to tumor purity (precisely defined) over a large number of bulk biological replicates and/or independent cases from the same type of tumor provides an efficient approach for identifying optimal transcriptional markers of malignant cells, since this strategy can sample billions of cells with a high signal-to-noise ratio. Furthermore, this strategy can be used to analyze other types of molecular features in parallel.

## Figures and Tables

**Figure 1 cancers-16-02429-f001:**
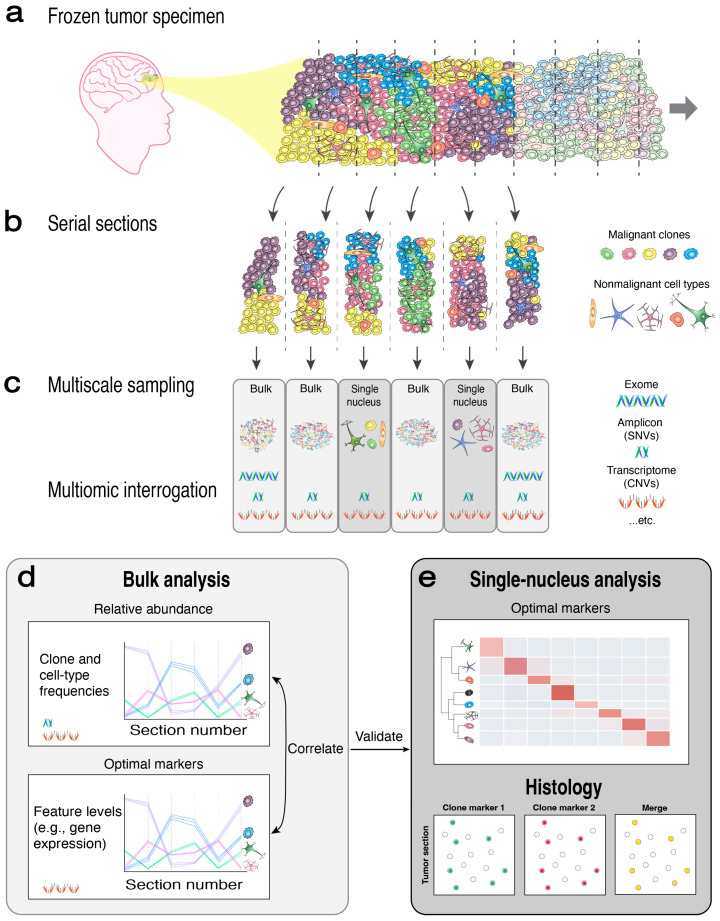
Overview of MOMA. (**a**) Schematic of a heterogeneous human brain tumor. (**b**) Serial sectioning introduces variation in cellular composition. (**c**) Section usage can be flexibly tailored for diverse multiscale and multiomic assays. (**d**) Correlative analysis of bulk cellular frequencies and molecular feature levels predicts optimal markers of malignant clones and nonmalignant cell types of the tumor microenvironment. (**e**) Predictions from bulk analysis are validated by single-nucleus analysis of interpolated sections and histology.

**Figure 2 cancers-16-02429-f002:**
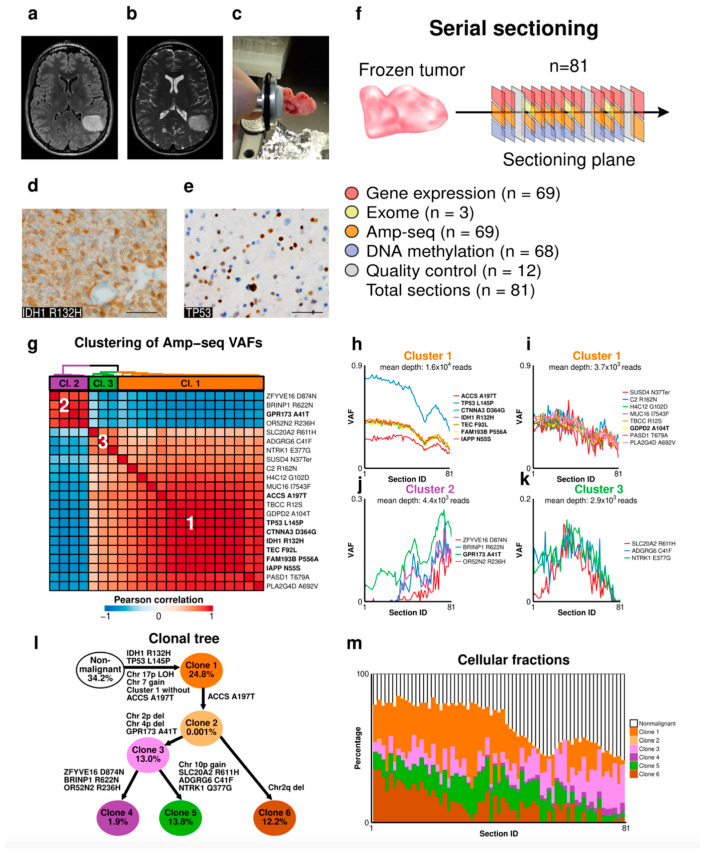
Multiomic analysis of serial tumor sections reveals the clonal composition of a primary grade 2 IDH-mutant astrocytoma (case 1). Axial T2 (**a**) and axial FLAIR (**b**) images demonstrate a round, well-defined T2 and FLAIR hyperintense intraaxial left temporoparietal mass that is non-enhancing and consistent with a low-grade glial neoplasm. (**c**) Image of the frozen tumor sample prior to cryosectioning and nucleic acid isolation. (**d**,**e**) Immunostaining for IDH1 R132H (**d**) and TP53 (**e**). Images: 400×. Scale bars: 50 μm. (**f**) Schematic of serial sectioning strategy and section usage plan. Amp-seq = deep sequencing of PCR amplicons spanning mutations identified by exome sequencing. (**g**) Hierarchical clustering of mutations, using 1—Pearson correlation of amp-seq variant allele frequencies (VAFs) over all tumor sections (n = 69) as a distance measure, reveals three clusters. Amp-seq was performed in two sequencing runs (denoted by bold and regular fonts). (**h**–**k**) VAF patterns comprising cluster 1 (**h**,**i**), cluster 2 (**j**), and cluster 3 (**k**). Cluster 1 was split to illustrate the effects of high (**h**) and low (**i**) coverage. (**l**) Clone phylogeny (with arbitrary branch lengths) derived from integrated analysis of SNVs (from amp-seq data) and CNVs (from DNA methylation data). Percentages represent the average abundance of each cellular fraction over all analyzed sections (n = 68). (**m**) Estimated cellular fractions for all clones and nonmalignant cells over all sections (n = 68).

**Figure 3 cancers-16-02429-f003:**
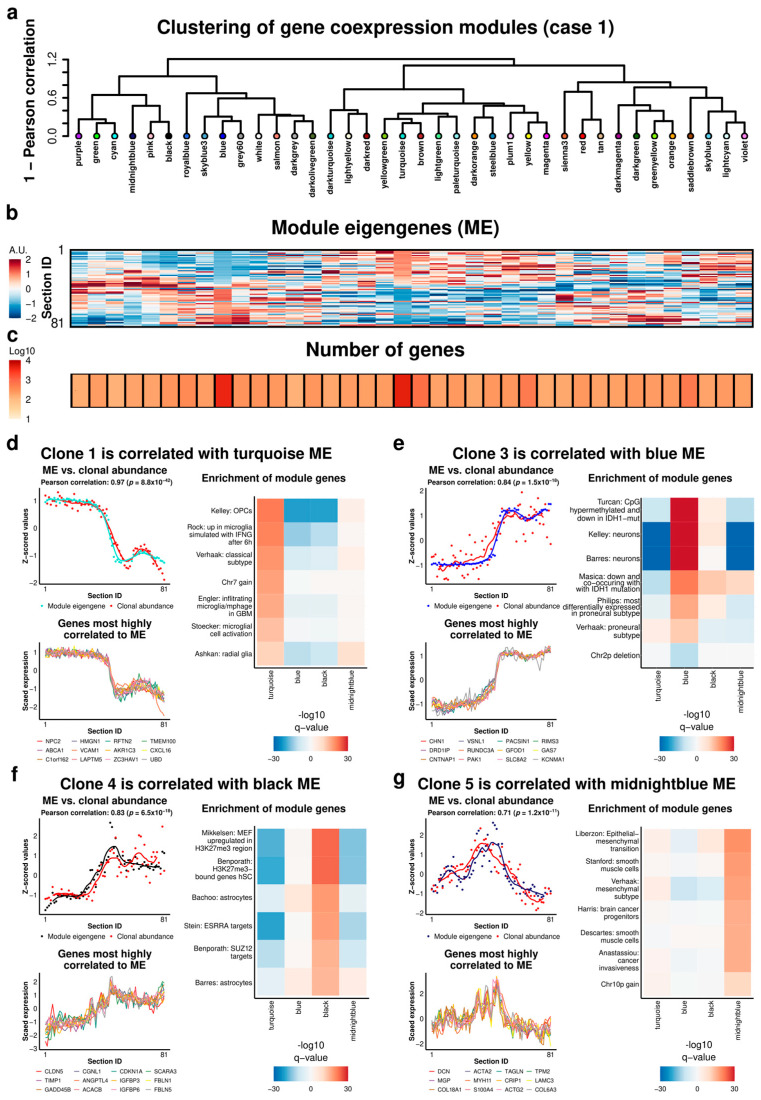
Gene coexpression modules are highly correlated with clonal abundance (case 1). (**a**) Hierarchical clustering of gene coexpression modules over all tumor sections (n = 69). (**b**) Module eigengenes (MEs) illustrate the relative expression levels of genes in each module over all tumor sections. (**c**) The number of genes used to form each ME. (**d**–**g**) Top left: MEs with the strongest correlations to clonal abundance (defined cumulatively). Locally weighted smoothing (LOESS) lines are shown; correlation is based on data points. Bottom left: the 12 genes with the highest correlations to the ME (*k*_ME_). Right: enrichment analysis of gene coexpression modules using published gene sets. FDR-corrected *p*-values (q-values) from one-sided Fisher’s exact tests are shown. Positive values represent enrichments of genes that were significantly positively correlated to the ME, while negative values represent enrichments of genes that were significantly negatively correlated to the ME. Gene sets representing chromosomal gains or losses include all genes within affected regions (as described in Figure 2l and Appendix A). See Appendix A for descriptions and sources of featured gene sets.

**Figure 4 cancers-16-02429-f004:**
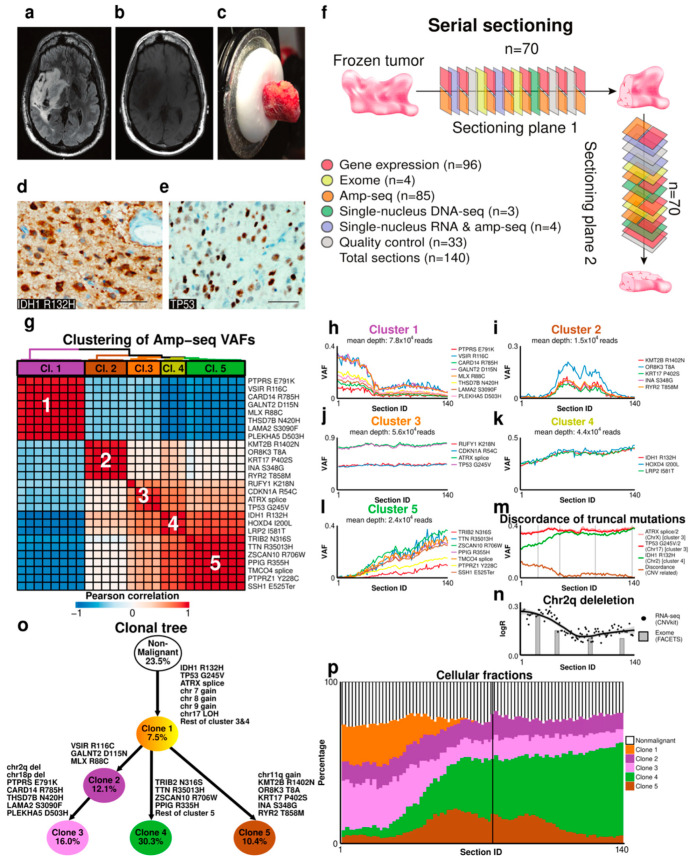
Multiomic analysis of serial tumor sections reveals the clonal composition of a recurrent grade 2 IDH-mutant astrocytoma (case 2). Axial T2 (**a**) and axial FLAIR (**b**) images demonstrate a non-enhancing, expansile, infiltrating glioma centered in the right insula and involving the basal ganglia, inferior frontal lobe, and temporal lobe. Cystic degeneration was present in the tumor. (**c**) Image of the frozen tumor specimen prior to cryosectioning and nucleic acid isolation. (**d**) The tumor was determined to harbor the IDH1 R132H mutation based on immunostaining with an antibody specific to the mutant protein. (**e**) TP53 immunostaining demonstrated nuclear expression with an estimated staining index of 20%. All histological images were captured at 400×. Scale bars denote 50 μm. (**f**) Schematic of serial sectioning strategy and section usage plan. (**g**) Hierarchical clustering of mutations, using 1—Pearson correlation of amp-seq VAFs over all tumor sections (n = 85) as a distance measure, revealing five clusters. (**h**–**l**) VAF patterns comprising cluster 1 (**h**), cluster 2 (**i**), cluster 3 (**j**), cluster 4 (**k**), and cluster 5 (**l**). (**m**) Controlling for gene dosage reveals discordance of IDH1 R132H VAF with respect to truncal *ATRX* and *TP53* mutations, which is explained by a subclonal deletion of chromosome 2q (including *IDH1*) that occurred after the *IDH1* point mutation. (**n**) Heatmap of the chromosome 2q deletion event frequency (as determined by FACETS [38]), with LOESS fit line (black) and smoothed 95% confidence interval (gray envelope). (**o**) Clone phylogeny (with arbitrary branch lengths) derived from integrated analysis of SNVs (from amp-seq data) and CNVs (from RNA-seq data). Percentages represent the average abundance of each cellular fraction over all analyzed sections (n = 85). (**p**) Estimated cellular fractions for all clones and nonmalignant cells over all sections. Black vertical line denotes orthogonal sample rotation.

**Figure 5 cancers-16-02429-f005:**
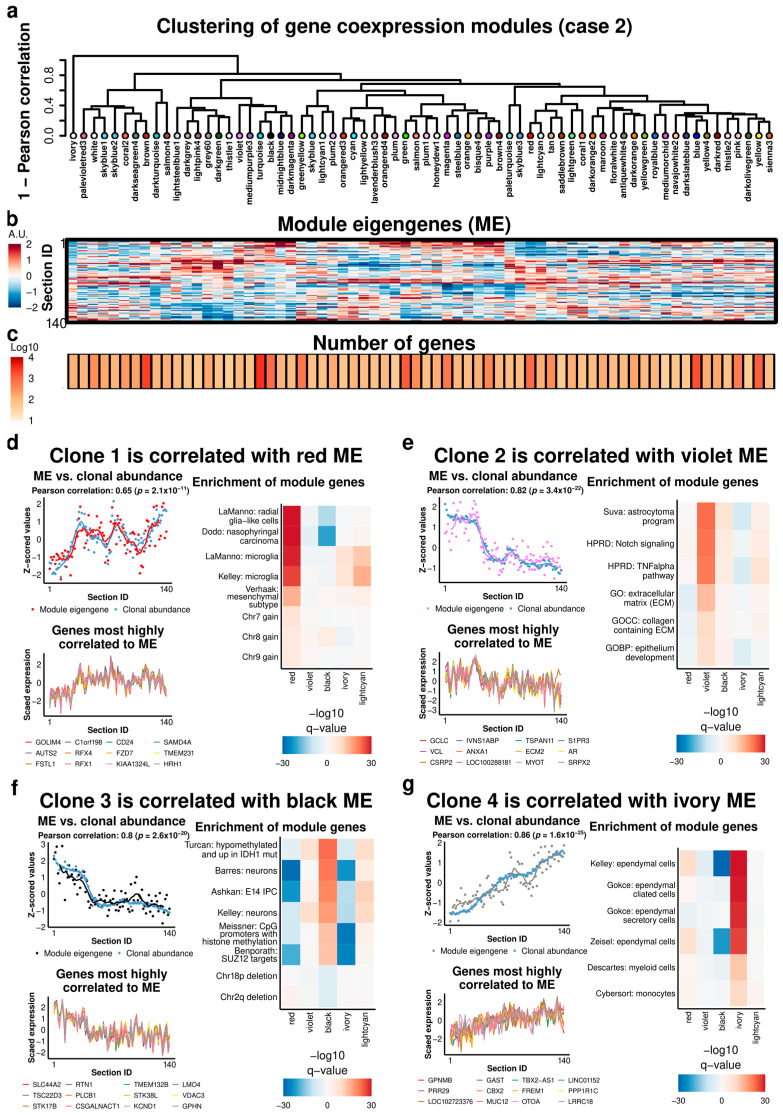
Gene coexpression modules are highly correlated with clonal abundance (case 2). (**a**) Hierarchical clustering of gene coexpression modules over all tumor sections (n = 90). (**b**) Module eigengenes (MEs) illustrate the relative expression levels of genes in each module over all tumor sections. (**c**) The number of genes that formed each ME. (**d**–**g**) Top left: MEs with the strongest correlations to clonal abundance (defined cumulatively). Locally weighted smoothing (LOESS) lines are shown; correlation is based on data points. Bottom left: the 12 genes with the highest correlations to the ME (*k*_ME_). Right: enrichment analysis of gene coexpression modules using published gene sets. FDR-corrected *p*-values (q-values) from one-sided Fisher’s exact tests are shown. Positive values represent enrichments of genes that were significantly positively correlated to the ME, while negative values represent enrichments of genes that were significantly negatively correlated to the ME. Gene sets representing chromosomal gains or losses include all genes within affected regions (as described in Figure 4o and Appendix A). See Appendix A for descriptions and sources of featured gene sets.

**Figure 6 cancers-16-02429-f006:**
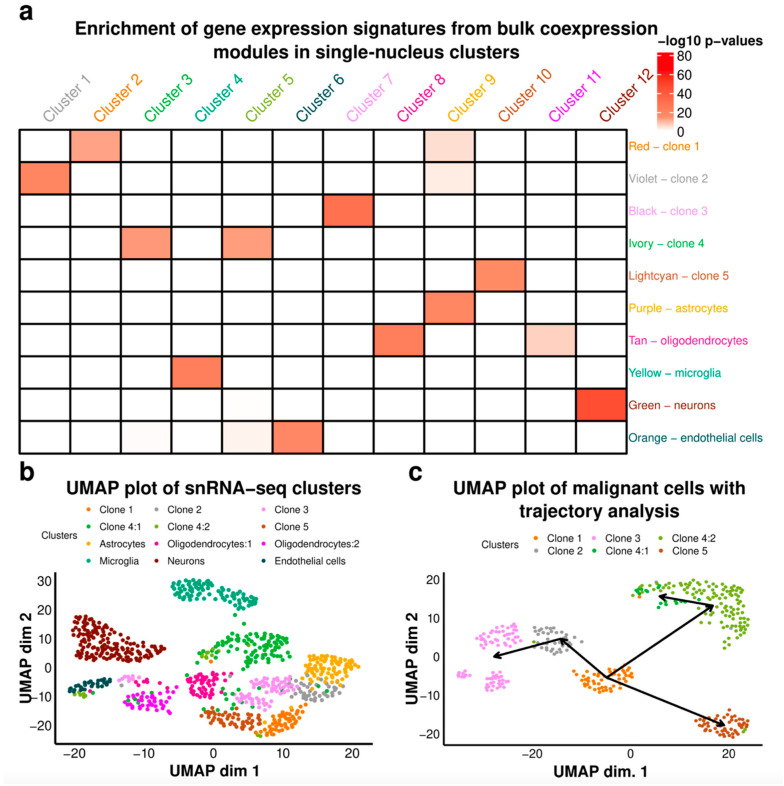
Single-nucleus RNA-seq analysis validates inferences from bulk data. (**a**) Heatmap of *p*-values (one-sided Wilcoxon rank-sum test) comparing differential expression t-values for genes comprising each bulk coexpression module (colors, x-axis) to all other genes in each SN cluster versus all other clusters. (**b**) UMAP plot of all nuclei (n = 809) with characterizations of clusters from (**a**) superimposed. (**c**) UMAP plot of malignant nuclei (n = 360), with results of Slingshot trajectory analysis [51] superimposed. Malignancy was determined by genotyping all nuclei via single-nucleus amplicon sequencing (snAmp-seq) of cDNA spanning mutations in the truncal clone.

**Figure 7 cancers-16-02429-f007:**
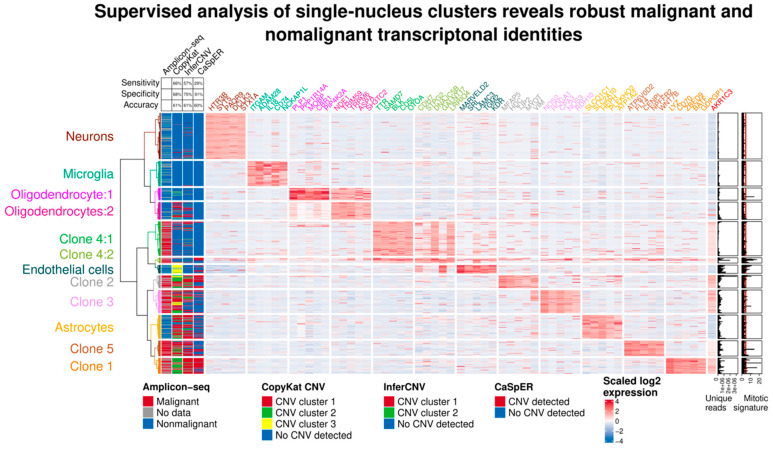
Genotyping nuclei profiled by snRNA-seq reveals the limitations of single-cell CNV-calling algorithms. Heatmap of scaled log_2_ expression vectors for the five most upregulated genes in each snRNA-seq cluster vs. all other clusters (one-sided Wilcoxon rank-sum test). Far left: malignancy vector determined by snAmp-seq of cDNA spanning mutations in the truncal clone. Left: malignancy vectors inferred from CNV analysis of snRNA-seq data using the CopyKat [49], InferCNV [25], or CaSpER [50] algorithms (blue = nonmalignant; all other colors = malignant). Right: bar plots depict the total number of unique reads (UMIs) for each nucleus and the average number of UMIs for genes comprising the Gene Ontology category ‘mitotic chromosome condensation’ (GO: 0030261). Red vertical line: max expression of mitotic genes in neurons, which presumably represents background noise.

**Figure 8 cancers-16-02429-f008:**
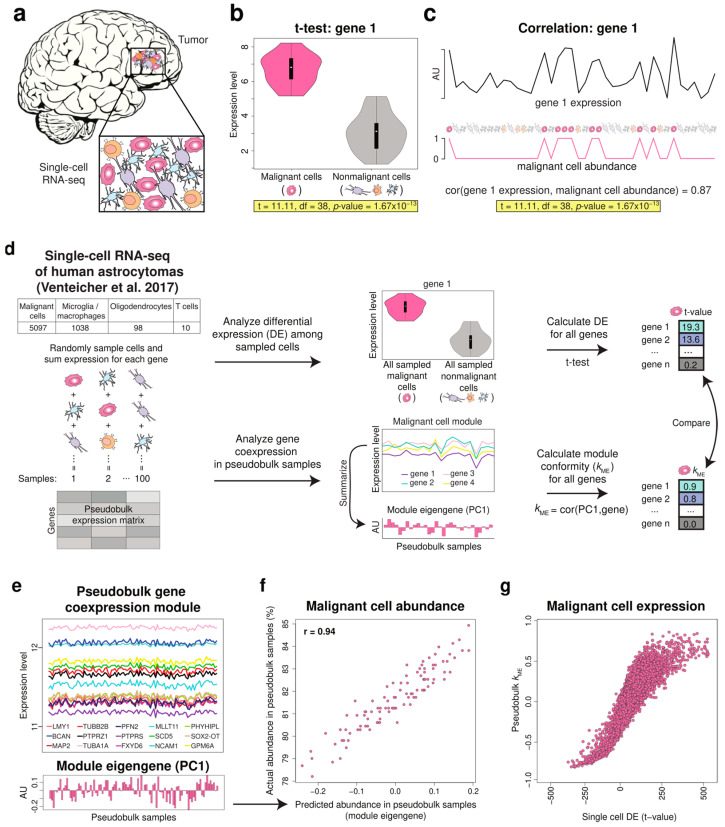
Correlation to malignant cell abundance predicts single-cell differential expression analysis of malignant vs. nonmalignant cells. (**a**–**d**) Analysis schematic. An adult malignant glioma consisting of malignant cells (pink) interspersed with nonmalignant cells (**a**). (**b**) Single-cell RNA-seq (scRNA-seq) reveals a hypothetical gene (gene *X*) that is significantly upregulated in malignant vs. nonmalignant cells. (**c**) Correlating the same gene’s expression pattern with a binary vector encoding malignant cell abundance (1 = malignant, 0 = nonmalignant) produces identical results. (**d**) Left: scRNA-seq data from 10 adult human IDH-mutant astrocytomas [24] were randomly sampled and aggregated to create 100 pseudobulk samples. Right (top): Genome-wide differential expression (DE) was analyzed for all sampled cells. Right (bottom): Genome-wide gene coexpression was analyzed for all pseudobulk samples. Each pseudobulk module was summarized by its module eigengene (PC1), which was compared to malignant cell abundance, and the correlation between each gene and each module eigengene (module conformity, or *k*_ME_) was calculated. (**e**) A pseudobulk malignant cell module featuring the top 15 genes ranked by *k*_ME_. By correlating the module eigengene to pseudobulk tumor purity (**f**), we see that this module is driven by variation in malignant cell abundance among pseudobulk samples. (**g**) The extent of DE (t-value) identified by scRNA-seq of malignant vs. nonmalignant cells predicts the correlation between gene expression and malignant cell abundance (pseudobulk *k*_ME_).

## Data Availability

All data are publicly available for download under NCBI BioProject ID PRJNA953039. Code for processing data and producing figures featured in this manuscript is available on GitHub: https://github.com/oldham-lab/Deconstructing-Intratumoral-Heterogeneity-through-Multiomic-and-Multiscale-Analysis.../tree/main (accessed on 26 June 2024).

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
