# Peer review of "Deconstructing Intratumoral Heterogeneity through Multiomic and Multiscale Analysis of Serial Sections"

_cancers, 2024, doi:10.3390/cancers16132429_

Round 1

Reviewer 1 Report

Comments and Suggestions for Authors

The authors describe a novel method, termed MOMA, developed to evaluate the clonal derivation of tumor cells in two IDH mutated gliomas. The study  in  corroborates known heterogeneity of tumor cells in gliomas shown in several pubications. The novel method, however, elegantly uses tissue slices from frozen tissue to investigate the spatial distribution of different clones.

Finally the authors discuss the advantages and limitations of the novel method.

Reviewer 2 Report

Comments and Suggestions for Authors

Intratumoral heterogeneity of brain cancers possesses a hurdle to targeted therapies.  Indeed, within the tumor there is a great spatial variation in spectrum and frequencies of pathogenic sequence variants. There is also a great spatial variation in spectrum and numbers of non-tumor cells within the tumor microenvironment (TME). Thus, there might be a significant difference between the sections of the same tumor. The authors based their study on the assumption (and observation) that there must be a “covariation of transcripts that are uniquely or predominantly expressed in specific kinds of cells” and, therefore, such molecular transcriptional patterns correlate with abundance of specific sub-populations (or clones) of cells within the TME. The authors propose to use multiomic and multiscale analysis (analysis of gene expression, whole exomes, deeply sequenced PCR amplicons spanning mutation sites, DNA methylation, and single-nucleus DNA and RNA) of serial tumor sections (MOMA) to address this issue, and provide a comprehensive set of data on application of such approach to analysis of IDH-mutant astrocytomas. Notably, the also claim that algorithms currently used for identifying cancer cells based on single-cell transcriptomes might be inaccurate.

The study is impressively comprehensive and “looks at the subject from different angles”. It is also well-written, although the writing style at times is too verbose and perceived as “too complex”.  I think that making the text more succinct and “simple” would have attracted greater audience, especially whose working outside of the field.

Do you think that analysis of circulating cell-free DNA (cfDNA) derived from tumor might address some of the aforementioned issues with tumor heterogeneity?

What is direct clinical (diagnostic) application of your work, especially in settings with limited resources/expertise?

Reviewer 3 Report

Comments and Suggestions for Authors

The authors' work as presented is high quality and deliberate in experimental design. While reasonable minds can disagree about bioinformatics approaches, there is no method deployed here that is objectionable--any disagreements I have on this front would be more preference and reflective of a philosophy of science than speaking to the rigor, which this work passes in abundance. I also believe that the conclusions are proportionate to the strength of the results, with any speculative statements clearly signified as such. Authors also cite and pull from well-respected works, even when not in full agreement with them. Overall a high quality work, clearly reflective of vast effort.

To my mind, there is only 1 aspect of the manuscript that needs addressing, though it should be relatively easy to perform.

1. Fig 6A. shows a heatmap of expression signature differences between clusters, but uses p value to do so. p values are thresholds of belief, not measures of effect. To be sure, there is a **general** relationship between the two, especially for small-to-modest sample sizes. However, at the scale of genes and scRNA data, that falls apart. The authors do not misrepresent the findings insofar as their main point is the (more-or-less) unique assignment of profile to cluster. However, it is bad form not to report an effect size generally and, for this paper in particular, does somewhat obfuscate the 'next-best' assignment (in truth, a minor issue given the strength of the results). Ideally, p value and effect would be shown together--however, if the authors think the message of the figure would be diminished by in-site inclusion, I think a supplementary table (referenced in-line) reporting the effect and p-val together is more than fine. The same general point can be made of figure S8. This is also an observation I have in some of the work around enrichment and annotation (e.g., the STRINGdb work). But the notion of effect size in these systems are less interpretable and, to my sensibilities, not more informative that p values. It is accepted practice to report only p value in those systems, as far as I can tell. 

Any other comments I make are more suggestions or considerations not dispositive (to my evaluation) to the publication worthiness of the document.

1.) In Fig 6, it is a bit disorienting that the color of the axis labels don't match with the cluster color they represent. Its doable and discernable without it, but if it is an easy change to sync colors across figures it'd help the readability of the paper.

2.) Not sure that it would show anything not shown in the work already, but a differential trajectory analysis could be useful to contrast malignant and other cell clusters (and is implemented in slingshot).

3.) I'm not worried about it so much in this work because there are multimodal results corroborating much of the observations made in them, but clone phylogenies can be very sensitive. PyClone is probably my top pick in software for doing them as well, but other programs make reasonable assumptions and can result in fairly different topologies and phylogenies. My rule of them is to proportion my claims (especially if they are topology reliant) to the meta-stability of the topology across software--even if I use PyClone for the basis of my clonal inference.

4. I do feel like there is a slight missed oppurtunity to do ancestral state reconstruction for the genes expression constituting the glioma subtypes to see if they had changed (i.e., shifted from one subtype to another) over progression. I've played with this in the GLASS data, but these serial sections add a degree of spacial resolution that improves temporal resolution that could make such an inference more informative, possibly revealing a path to and between subtypes.

The sentence starting: "Third, we compared estimates of cellular abun-
dance obtained..." on (438-441) is in a different font than the rest of the document.
